# A FRAMEWORK OF DEEP NEURAL NETWORKS VIA THE SOLUTION OPERATOR OF PARTIAL DIFFERENTIAL EQUATIONS

## ABSTRACT

There is a close connection between deep neural networks (DNN) and partial differential equations (PDEs). Many DNN architectures can be modeled by PDEs and have been proposed in the literature. However, their neural network design space is restricted due to the specific form of PDEs, which prevents the design of more effective neural network structures. In this paper, we attempt to derive a general form of PDEs for the design of ResNet-like DNN. To achieve this goal, we first formulate DNN as an adjustment operator applied on the base classifier. Then based on several reasonable assumptions, we show the adjustment operator for ResNet-like DNN is the solution operator of PDEs. To show the effectiveness of the general form of PDEs, we show that several effective networks can be interpreted by our general form of PDEs and design a training method motivated by PDEs theory to train DNN models for better robustness and less chance of overfitting. Theoretically, we prove that the robustness of DNN trained with our method is certifiable and our training method reduces the generalization gap for DNN. Furthermore, we demonstrate that DNN trained with our method can achieve better generalization and is more resistant to adversarial perturbations than the baseline model.

## 1 INTRODUCTION

Deep neural networks (DNN) have achieved success in tasks such as image classification (Simonyan & Zisserman, 2015), speech recognition (Dahl et al., 2012), video analysis (Bo et al., 2011), and action recognition (Wang et al., 2016). Residual learning (He et al., 2016a;b) has revolutionized DNN architecture design, making it practical to train ultra-deep DNN, and has ability to avoid gradient vanishing. The idea of residual learning has motivated the development of powerful DNN such as WideResNet (Zagoruyko & Komodakis, 2016), ResNeXt (Xie et al., 2017), and DenseNet (Huang et al., 2017).

So how to understand Residual learning from a theoretical perspective? In (Weinan, 2017), the author formulated the forward propagation of ResNet as a dynamic system and in (Li & Shi, 2017; Haber et al., 2018; Li et al., 2017) bridge the connection between partial differential equations(PDE) and ResNet. In fact, ResNet is composed of a series of residual mapping. Each residual block is realized by adding a shortcut connection to connect the input and output of the original mapping ($\mathcal{F}$). Mathematically, the $n$-th residual mapping can be formulated as

$$\boldsymbol{x}_{n+1} = \boldsymbol{x}_n + \mathcal{F}(\boldsymbol{x}_n, \boldsymbol{w}_n) \tag{1}$$

with $\boldsymbol{x}_0 \in \mathbb{R}^d$ be a data point. $\boldsymbol{x}_n$ is the input to the $n$-th residual mapping and $\mathcal{F}(\boldsymbol{x}_n, \boldsymbol{w}_n)$ is a non-linear function parameterized by $\boldsymbol{w}_n$ which can be learned by back propagating the training error. The prediction of the data point $\boldsymbol{x}_0$ can be expressed by $\hat{y} = \mathrm{softmax}(\boldsymbol{w}^{\mathrm{FC}}\boldsymbol{x}_L)$. By introducing a temporal partition: $t_n = nT/L$, for $n = 0, 1, .., L$ with the time interval $\Delta t = T/L$ and let $\boldsymbol{x}(t_n) = \boldsymbol{x}_n$ and $\boldsymbol{w}(t_n) = \boldsymbol{w}_n$. Without considering the dimensional consistency, ResNet can be formulated as :

$$\boldsymbol{x}(t_{n+1}) = \boldsymbol{x}(t_n) + \frac{\mathcal{F}(\boldsymbol{x}(t_n), \boldsymbol{w}(t_n))}{\Delta t} \Delta t, n = 0, 1, \cdots, L,$$

which is the forward Euler discretization of the following equations with time step $\Delta t$, that is,

$$d\boldsymbol{x}(t) = v(\boldsymbol{x}(t), t)dt, \ \boldsymbol{x}(0) = \boldsymbol{x}, \ \ t \in [0, T], \tag{2}$$

where $v(\boldsymbol{x}(t_n), t_n) = \mathcal{F}(\boldsymbol{x}(t_n), \boldsymbol{w}(t_n))/\Delta t$ and $L$ is the number of resdiual mapping in a ResNet architecture. The prediction can be expressed by $\hat{y} = \text{softmax}(\boldsymbol{w}^{\text{FC}}\boldsymbol{x}(T))$. The equation 2 defines the characteristic curves of the following TE

$$\frac{\partial u}{\partial t}(\boldsymbol{x}, t) = v(\boldsymbol{x}, t)\nabla u(\boldsymbol{x}, t), \ \boldsymbol{x} \in \mathbb{R}^d. \tag{3}$$

So the prediction $\hat{y}$ of ResNet can be seen as the solution $u(\boldsymbol{x}, T)$ of following Transport equation(TE):

$$\begin{cases} \frac{\partial u}{\partial t}(\boldsymbol{x}, t) = v(\boldsymbol{x}, t)\nabla u(\boldsymbol{x}, t), \ x \in \mathbb{R}^d. \\ u(\boldsymbol{x}, 0) = \text{softmax}(\boldsymbol{w}^{\text{FC}}\boldsymbol{x}). \end{cases} \tag{4}$$

In fact, not only ResNet, similar to the process above, many ResNet-like DNN can correspond to PDEs. For example, it has been shown (Lu et al., 2018) that PolyNet (Zhang et al., 2017) and FractalNet (Larsson et al., 2016) can be respectively considered numerical schemes to solve a PDEs system. Based on the link of PDEs and ResNet-like DNN, many theoretical results to understand ResNet-like DNN were proved. In (Chen et al., 2018), a continuous form for ResNet called Neural ODE was proposed. After that, the property of Neural ODE has been theoretically analyzed (Thorpe & van Gennip, 2018; Avelin & Nyström, 2021; Zhang et al., 2019a). Based on the stability theory of differential equations, many architectures has been designed to build more robust deep networks (Zhang et al., 2019b; Yan et al., 2019; Reshniak & Webster, 2021). Not only stability theory, inspired by the numerical scheme of differential equations, in (Zhu et al., 2018; Tao et al., 2018), various deep network architectures to improve testing accuracy have been proposed in success.

Despite the success of the application of PDEs in residual learning, these neural network design space in the literature mentioned above is restricted due to the specific form of PDEs, which prevents the design of more effective neural network structures. Existed works can not meet these requirements. So, a natural question is that can we find the general form of the PDEs corresponding to the ResNet-like DNN which can unify existed model as well as guide new effective DNN architecture design?

To achieve this goal, in this paper, we follow the setting of paragraph 2, the softmax model and ResNet are the initial value and the terminal value of PDEs equation 4 respectively, so we can view ResNet as an evolution from a softmax model. Without loss of generality, we indicate this evolutive relationship by an adjustment operator denoted by $T_t$, then we have

$$T_t : \text{softmax}(\boldsymbol{w}^{FC}\boldsymbol{x}) \rightarrow \text{ResNet}(\boldsymbol{x}).$$

If we denote the softmax model by $f$, then the ResNet-like DNN can be expressed as $T_t(f)$ and $T_t$ is the solution operator of PDEs equation 4. To obtain the general form of the PDEs corresponding to these ResNet-like DNN, we list a series of formal properties likely to be satisfied by the operator $T_t$, and we deduce an explicit formulation from them. We see that under reasonable assumptions, the new model $u(\boldsymbol{x}, t) = (T_t f)(\boldsymbol{x})$ is the solution of convection diffusion equations. Then, motivated by the theory of PDEs and the framework of our general form of PDEs, we design a training method for ResNet-like DNN models for better robustness and less overfitting risk. Our training method is summarized as follows. First, we train a ResNet-like DNN model using natural training. Because calculating the real solution $T_t(f)(\boldsymbol{x})$ is difficult, we next train a new ResNet-like DNN denoted by $g(\boldsymbol{x}, t)$ to approximate $T_t(f)(\boldsymbol{x})$. At last, we choose $g(\boldsymbol{x}, T)$ as our final classifier.

We make the following contributions:

- Under reasonable assumptions, we build a PDEs framework for ResNet-like DNN with the help of operator $T_t$ and show some effective models that can be modeled by our framework.

- We design a training method for DNN based on the theory of PDEs. This is an essential contribution of this paper due to certified adversarial robustness and theoretical analysis of generalization ability for our training method.

- We verify our training method on a synthetic dataset and the CIFAR-10/CIFAR-100 datasets, then demonstrate that a ResNet trained by our method is more resistant to multistep $\ell^2$ and $\ell^\infty$ adversarial attacks than natural trained ResNet.

## 2   MAIN RESULTS

We show under several reasonable assumptions that the adjustment operator $T_t$ is the solution operator of convection diffusion equations. Based on the regularity theory of PDEs, we design a training method for DNN models and theoretically illustrate its benefits for improving the robustness of models and reducing the generalization gap.

### 2.1   GENERAL FORM OF PDES

We denote the base DNN model by $f_{\boldsymbol{w}}(\boldsymbol{x})$, which is uniformly continuous. We assume data points are distributed in $\mathcal{D} \subset \mathbb{R}^d$, and label function $l(\boldsymbol{x})$ is defined in $\mathcal{D}$ but known only in training set $S_N \in \mathcal{D}$. Our method formulates the DNN as a continuous dynamic system, namely $u(\boldsymbol{x}, t) = T_t(f), t \in [0, T]$. To get the expression of the adjustment operator $T_t$, we assume it has some fundamental properties.

**[Locality]** $(T_t(f) - T_t(g))(\boldsymbol{x}) = o(t)$ as $t \to 0^+$, for all $f, g$ such that $D^\alpha f(\boldsymbol{x}) = D^\alpha g(\boldsymbol{x})$ for all $|\alpha| \geq 0$.

**[Comparison Principle]** $T_t(f) \leq T_t(g)$ for all $t \geq 0$ and $f, g$ such that $f \leq g$.

**[Markov Property]** $T_{t+s} = T_{t+s,s} \circ T_s$, for all $s, t \geq 0$ and $t + s \leq T$. $T_{t+s,s}$ denotes the flow from time $s$ to time $t + s$.

**[Spatial Regularity]** There exist a positive constant $C$ such that $\|T_t(\tau_{\boldsymbol{h}} f) - \tau_{\boldsymbol{h}}(T_t f)\|_{L^\infty} \leq Ch$ for all $\boldsymbol{h}$ in $\mathbb{R}^d$, $t \geq 0$, where $(\tau_{\boldsymbol{h}} f)(\boldsymbol{x}) = f(\boldsymbol{x} + \boldsymbol{h})$ and $\|\boldsymbol{h}\|_2 = h$.

**[Temporal Regularity]** The following two properties hold:
1) For all $t, s, t + s \in [0, T]$ and all $g \in Q$, there exist a constant $C \geq 0$ depending only on $Q$ such that $\|T_{t+s,s}(g) - g\|_{L^\infty} \leq Ct$, where $Q$ is the subset of $C_b^\infty$ defined by

$$Q = \{f \in C_b^\infty, \forall n \geq 0, \|D^\alpha f\|_{L^\infty} \leq C \text{ for all } |\alpha| = n\}.$$

2) For all $t, s, t + s \in [0, T]$, there exist a constant $C \geq 0$ such that $\|T_{t+s,t}(f) - T_{s,0}(f)\|_{L^\infty} \leq Cst$ uniformly in $s \in (0, T]$.

**[Ensemble Additivity]** $T_t(f + g) = T_t(f) + T_t(g)$, and if $C$ is a constant, then $T_t(C) = C$.

To illustrate the reasonableness of these assumptions above, we give the following explanation. We first require that $T_t$ can be expressed as a solution operator of PDEs and for small enough $t$, the value of $T_t(f)$ at any point $\boldsymbol{x}$ is determined by the behavior of the base classifier $f$ near $\boldsymbol{x}$, which implies **[Locality]**. We then require the operator $T_t$ is provided with an order-preserving property which means that no enhancement is made. Thus if the confidence level for an event of one base classifier $g$ is higher than another base classifier $f$, this ordering is preserved, which implies **[Comparison Principle]**. Next we require $T_{t+s}$ can be computed from $T_t$ for any $s \leq t$, and $T_0$ is of course the identity. This is natural, since the future behaviour of the adjustment process at terminal time $t + s$ is likely to be deduced from the state at intermediate time $t$ without any dependence upon the original model $f$, which implies **[Markov Property]**. We need function $T_t(f)$ to be spatial stable. If we add a perturbation $\boldsymbol{h}$ to data point $\boldsymbol{x}$, the output $T_t(f)(\boldsymbol{x} + \boldsymbol{h})$ will not change a lot, which implies **[Spatial Regularity]**. We also need an additional property for giving some temporal stability to the operator $T_t$. Namely, in any time interval, the adjustment will not be rapid, which implies **[Temporal Regularity]**. Assume an ensemble classifier $F$ consists of base classifiers $f$ and $g$, and another ensemble classifier consists of base classifiers $T_t(f)$ and $T_t(g)$. We require the prediction $T_t(F)$ to equal $T_t(f) + T_t(g)$, which implies **[Ensemble Additivity]**.

**Theorem 1.** *Under the above conditions, there exists Lipschitz continuous function $v : \mathbb{R}^d \times [0, T] \to \mathbb{R}^d$ and Lipschitz continuous positive function $\sigma : \mathbb{R}^d \times [0, T] \to \mathbb{R}^{d \times d}$ such that for any bounded and uniformly continuous base model $f_{\boldsymbol{w}}(\boldsymbol{x})$, $u(\boldsymbol{x}, t) = T_t(f_{\boldsymbol{w}})(\boldsymbol{x})$ is the unique viscosity solution of the following convection diffusion equation:*

$$\begin{cases} \frac{\partial u(\boldsymbol{x},t)}{\partial t} = v(\boldsymbol{x}, t) \cdot \nabla u(\boldsymbol{x}, t) + \frac{1}{2} \sum_{i,j} \sigma_{i,j} \frac{\partial^2 u}{\partial x_i \partial x_j}(\boldsymbol{x}, t), \\ u(\boldsymbol{x}, 0) = f_{\boldsymbol{w}}(\boldsymbol{x}), \end{cases} \quad (5)$$

*where $\boldsymbol{x} \in \mathbb{R}^d, t \in [0, T]$. Here $\sigma_{i,j}$ is the $i, j$-th element of matrix function $\sigma(\boldsymbol{x}, t)$.*

We will provide the proof of Theorem 1 in Appendix A. In this subsection, we introduce a PDEs framework for DNN models. The framework is quite general, many existing models with residual connections can be modeled by our framework. As examples, we briefly list some of them in the next subsection.

## 2.2 SEVERAL MODELS

In this subsection, we will show Gaussian noise injection(Wang et al., 2020; Liu et al., 2019), randomized smoothing(Cohen et al., 2019; Li et al., 2018; Salman et al., 2019) and ResNet with stochastic dropping out the hidden state of residual block(Srivastava et al., 2014; Sun et al., 2018) can be interpreted by our framework.

**Gaussian noise injection:** Gaussian noise injection is an effective regularization mechanism for a DNN model. For a base ResNet with $L$ residual mapping, the $n$-th residual mapping with Gaussian noise injected can be written as

$$\boldsymbol{x}_{n+1} = \boldsymbol{x}_n + \mathcal{F}(\boldsymbol{x}_n, \boldsymbol{w}_n) + a\boldsymbol{\varepsilon}_n, \ \ \boldsymbol{\varepsilon}_n \sim \mathcal{N}(0, \mathbf{I}),$$

where the parameter $a$ is a noise coefficient. By introducing a temporal partition: $t_n = nT/L$, for $n = 0, 1, .., L$ with the time interval $\Delta t = T/L$ and let $\boldsymbol{x}(t_n) = \boldsymbol{x}_n$ and $\boldsymbol{w}(t_n) = \boldsymbol{w}_n$. And let $a = \sigma\sqrt{\Delta t}$ and $\mathcal{F}(\boldsymbol{x}_n, \boldsymbol{w}_n)/\Delta t = v(\boldsymbol{x}, t)$. This noise injection technique in a discrete neural network can be viewed as the approximation of continuous dynamic

$$d\boldsymbol{x}(t) = v(\boldsymbol{x}_n, t)dt + \sigma d\mathbf{B}(t) \tag{6}$$

where $\mathbf{B}(t)$ is multidimensional Brownian motion. The output of $L$-th residual mapping can be written as Itô process equation 6 at terminal time $T$, $\boldsymbol{x}(T)$. So, an ensemble prediction over all the possible sub-networks with shared parameters can be written as

$$\hat{y} = \mathbb{E}(\text{softmax}(\boldsymbol{w}^{\text{FC}}\boldsymbol{x}(T))|\boldsymbol{x}(0) = \boldsymbol{x}_0). \tag{7}$$

According to Feynman-Kac formula(Mao, 2007), equation equation 7 is known to solve the following convection diffusion equation

$$\begin{cases} \frac{\partial u(\boldsymbol{x},t)}{\partial t} = v(\boldsymbol{x}, t) \cdot \nabla u + \frac{1}{2}\sigma^2\Delta u, \boldsymbol{x} \in \mathbb{R}^d, t \in [0, T] \\ u(\boldsymbol{x}, 0) = \text{softmax}(\boldsymbol{w}^{\text{FC}}\boldsymbol{x}) \end{cases}$$

**Randomized Smoothing:** Consider to transform a base classifier into a new smoothed classifier by adding Gaussian noise to the input when inference time. If we denote the base classifier by $f(\boldsymbol{x})$ and denote the new smoothed classifier by $g(\boldsymbol{x})$. Then $f(\boldsymbol{x})$ and $g(\boldsymbol{x})$ have the following relation:

$$g(\boldsymbol{x}) = \frac{1}{N}\sum_{i=1}^{N} f(\boldsymbol{x} + \boldsymbol{\varepsilon}_i) \approx \mathbb{E}_{\boldsymbol{\varepsilon}\sim\mathcal{N}(0,I)}[f(\boldsymbol{x} + \boldsymbol{\varepsilon})] \tag{8}$$

According to Feynman-Kac formula, $g(\boldsymbol{x})$ can be viewed as the solution of the following PDEs

$$\begin{cases} \frac{\partial u(\boldsymbol{x},t)}{\partial t} = \frac{1}{2}\Delta u, t \in [0, 1] \\ u(\boldsymbol{x}, 0) = f(\boldsymbol{x}). \end{cases} \tag{9}$$

Especially, when $f(\boldsymbol{x})$ is ResNet, the smoothed classifier $g(\boldsymbol{x}) = u(\boldsymbol{x}, T + 1)$ can be expressed as

$$\begin{cases} \frac{\partial u(\boldsymbol{x},t)}{\partial t} = \frac{1}{2}\Delta u, t \in [T, T + 1] \\ \frac{\partial u(\boldsymbol{x},t)}{\partial t} = v(\boldsymbol{x}, t) \cdot \nabla u(\boldsymbol{x}, t), \boldsymbol{x} \in \mathbb{R}^d, t \in [0, T] \\ u(\boldsymbol{x}, 0) = \text{softmax}(\boldsymbol{w}^{\text{FC}}\boldsymbol{x}). \end{cases}$$

**Dropout of Hidden Units:** Consider the case that disables every hidden units independently from a Bernoulli distribution $\mathcal{B}(p)$ with $p \in (0, 1)$ in each residual mapping

$$\boldsymbol{x}_{n+1} = \boldsymbol{x}_n + \mathcal{F}(\boldsymbol{x}_n, \boldsymbol{w}_n) \odot \frac{\boldsymbol{z}_n}{p}$$

$$= \boldsymbol{x}_n + \mathcal{F}(\boldsymbol{x}_n, \boldsymbol{w}_n) + \mathcal{F}(\boldsymbol{x}_n, \boldsymbol{w}_n) \odot (\frac{\boldsymbol{z}_n}{p} - \mathbf{I}),$$

where $z_n \sim \mathcal{B}(1, p)$ namely $\mathbb{P}(z_n = 0) = 1 - p$, $\mathbb{P}(z_n = 1) = p$ and $\odot$ indicates the Hadamard product. If the number of the ensemble is large enough, according to Central Limit Theorem, we have

$$\mathcal{F}(\boldsymbol{x}_n, \boldsymbol{w}_n) \odot (\frac{\boldsymbol{z}_n}{p} - \mathbf{I}) \approx \mathcal{F}(\boldsymbol{x}_n, \boldsymbol{w}_n) \odot \mathcal{N}(0, \frac{1-p}{p}).$$

The similar way with Gaussian noise injection, the ensemble prediction $\hat{y}$ can be viewed as the solution $u(\boldsymbol{x}, T)$ of following equation:

$$\begin{cases} \frac{\partial u(\boldsymbol{x},t)}{\partial t} = v(\boldsymbol{x}, t) \cdot \nabla u(\boldsymbol{x}, t) + \frac{1-p}{2p} \sum_i (v^T v)_{i,i} \frac{\partial^2 u}{\partial x_i^2}(\boldsymbol{x}, t), \\ u(\boldsymbol{x}, 0) = \text{softmax}(\boldsymbol{w}^{\text{FC}} \boldsymbol{x}) \end{cases} \tag{10}$$

where $\boldsymbol{x} \in \mathbb{R}^d, t \in [0, T]$.

**Remark 1.** *In fact, similar to dropout, shake-shake regularization (Gastaldi, 2017; Huang & Narayanan, 2018) and ResNet with stochastic depth (Huang et al., 2016) can also be formulated by our PDEs framework.*

The viewpoint that forward propagation of a ResNet is the solving process of a TE enables us to interpret poor robustness and generalization ability as the irregularity of the solution. Morever we can also interpret the effectiveness of the models above-mentioned in improving generalization as well as robustness as the action of diffusion term. To mitigate this issue, based on our PDEs framework and the knowledge that the terminal value of convection diffusion equations (whose diffusion term is isotropic, namely $\sigma(\boldsymbol{x}, t)$ is $\sigma^2 \mathbf{I}$) is more regular than initial value, we use the ResNet $f_{\boldsymbol{w}}(\boldsymbol{x})$ as the base classifier and use the solution operator of the diffusion equation

$$\begin{cases} \frac{\partial u(\boldsymbol{x},t)}{\partial t} = v(\boldsymbol{x}, t) \cdot \nabla u(\boldsymbol{x}, t) + \sigma^2 \Delta u, \boldsymbol{x} \in \mathbb{R}^d, t \in [0, T] \\ u(\boldsymbol{x}, 0) = f_{\boldsymbol{w}}(\boldsymbol{x}). \end{cases} \tag{11}$$

as the adjustment operator $T_t$ to smooth the ResNet for improving the robustness and generalization ability. The case that diffusion term $\sigma(\boldsymbol{x}, t)$ is anisotropic (namely $\sigma(\boldsymbol{x}, t) \neq \sigma^2 \mathbf{I}$) is difficult for theoretical analysis and practical experiment and we take the case as our future work. In next subsection, we will illustrate the benefits of using the adjustment operator $T_t$ from a theoretical perspective.

## 2.3 ROBUSTNESS GUARANTEE

Consider a classification problem from $\mathbb{R}^d$ to blue label class $\mathcal{Y}$. Let $G$ be a classifier defined by $G(\boldsymbol{x}) = \arg\max_{i \in \mathcal{Y}} u^i(\boldsymbol{x}, T)$, where $u^i(\boldsymbol{x}, T)$ is the $i$-th element of $u(\boldsymbol{x}, T)$. Suppose that the new DNN classifies $\boldsymbol{x}$ the most probable class $c_A$ is returned with probability $p_A = u^{c_A}(\boldsymbol{x}, T)$, and the "runner-up" class is returned with probability $p_B$. Our main result of this subsection is to estimate around the data point $\boldsymbol{x}$ how large $\ell^p$ radius can make the classifier $G(\boldsymbol{x})$ be robust, where $1 < p \leq \infty$.

**Theorem 2.** *Let the base model $f_{\boldsymbol{w}}(\boldsymbol{x})$ be a compactly supported function and velocity $v(\boldsymbol{x}, t) \in C^1(\mathbb{R}^d \times [0, T])$. Let the new DNN $u(\boldsymbol{x}, T)$ be the solution of equation 11. Suppose $c_A \in \mathcal{Y}$ and*

$$u^{c_A}(\boldsymbol{x}, T) = p_A \geq p_B = \max_{i \neq c_A} u^i(\boldsymbol{x}, T),$$

*then $G(\boldsymbol{x} + \boldsymbol{\delta}) = c_A$ for all $\|\boldsymbol{\delta}\|_p \leq R$, where*

$$R = \frac{e^{\sigma^2 T}(p_A - p_B)}{2d^{1/q} e^{\gamma T} \max_i \|f_{\boldsymbol{w}}^i(\boldsymbol{x})\|_{C^1}},$$

*$1/p + 1/q = 1$ and $\gamma$ is a constant depending on $\nabla v$.*

We provide the proof of Theorem 2 in Appendix B. According to Theorem 2, we can obtain the certified radius $R$ is large when the diffusion coefficient $\sigma^2$ and the probability of the top class is high. Not only robustness, comparing to the base classifier, but we will also show the generalization ability of model 11 will be better in next subsection.

### 2.4 GENERALIZATION GAP

For simplicity, we consider binary classification problems. Suppose $S_N = \{(\boldsymbol{x}_i, y_i)\}_{i=1}^N$ is drawn from $X \times Y \subset \mathcal{D} \times \{-1, +1\}$ with $X$ and $Y$ being the input data and label spaces, respectively. Assume $p(\boldsymbol{x})$ is the underlying distribution of $X$, which is defined in $\mathcal{D}$. We also assume label function $\text{Label} : \mathcal{D} \to \{-1, +1\}$, which is unknown. Let $\mathcal{H} \subset V^X$ be the hypothesis class of the DNN model, where $V$ is another space that might be different from $Y$. Let $\ell : V \times Y \to [0, B]$ be the loss function and $B$ is a positive constant. Denote function class $\ell_{\mathcal{H}} := \{(\boldsymbol{x}, y) \to \ell(h(\boldsymbol{x}), y) : h \in \mathcal{H}\}$. Rademacher complexity is one of the classic measures of generalization error. We first recap on the definition of Rademacher complexity.

**Definition 1.** *(Ledoux & Talagrand, 2013)Let $\mathcal{H} : X \to \mathbb{R}$ be the space of real-valued functions on the space $X$. For a given sample $S_N = \{(\boldsymbol{x}_i, y_i)\}_{i=1}^N$, the empirical Rademacher complexity of $\mathcal{H}$ is defined as*

$$R_{S_N}(\mathcal{H}) := \frac{1}{N}\mathbb{E}_\sigma[\sup_{h \in \mathcal{H}} \sum_{i=1}^N \sigma_i h(\boldsymbol{x}_i)],$$

*where $\sigma_1, \sigma_2, \cdots, \sigma_N$ are i.i.d. Rademacher random variables with $\mathbb{P}(\sigma_i = 1) = \mathbb{P}(\sigma_i = -1) = \frac{1}{2}$.*

Rademacher complexity is a tool to bound the generalization error. The smaller the generalization gap is, the less overfitting the model is. For $\forall \boldsymbol{x}_i \in \mathbb{R}^d$ and constant $c \geq 0$, we consider the following two function classes:

$$\mathcal{F} := \Big\{f(\boldsymbol{x})|\|f\|_{C^1} \leq c\Big\},$$

and

$$\mathcal{G}_\sigma := \Big\{g(\boldsymbol{x}) = u(\boldsymbol{x}, 1)|\frac{\partial u(\boldsymbol{x}, t)}{\partial t} = v(\boldsymbol{x}, t)\nabla u(\boldsymbol{x}, t) + \sigma^2 \Delta u, u(\boldsymbol{x}, 0) = f(\boldsymbol{x}) \text{ where } f \in \mathcal{F}\Big\}$$

The function class $\mathcal{F}$ represents the continuous analogue of ResNet, and function class $\mathcal{G}_\sigma$ denotes $\mathcal{F}$ adjusted by operator $T_t$. Then we have the following theorem.

**Theorem 3.** *Given a training set $S_N = \{(\boldsymbol{x}_i, y_i)\}_{i=1}^N$. With probability at least $1 - 1/(2N)$ we have*

$$R_{S_N}(\mathcal{F}) \leq \frac{C}{\sqrt{N}}[(\log c + \log N)^{1/2} + c]$$

$$R_{S_N}(\mathcal{G}_\sigma) \leq \frac{C}{\sqrt{N}}[(-\sigma^2 + \ln c + \ln N)^{1/2} + \exp(-\sigma^2)c]$$

*Here $C$ is a positive constant depend on $\mathcal{D}$ and dimension $d$.*

We provide the proof of Theorem 3 in Appendix C. According to Theorem 3, we can obtain that for function class $\mathcal{G}_\sigma$, the upper bound of Rademacher complexity is low when the diffusion coefficient $\sigma^2$ is high. We next present a practical algorithm for real applications to verify our main results.

## 3 ALGORITHMS

Let $S_N = \{(\boldsymbol{x}_1, y_1), \cdots, (\boldsymbol{x}_n, y_n)\}$ be training set, where $\boldsymbol{x}_i$ is a data point and $y_i$ is the corresponding label. Then denote the ResNet with natural training by $f_{\boldsymbol{w}}(\boldsymbol{x})$ whose parameters can be learned by minimized the loss $l^{CE}(y_i, f_{\boldsymbol{w}}(\boldsymbol{x}_i))$, where $l^{CE}$ is cross-entropy loss. . For convenience, we assume the velocity $v(\boldsymbol{x}, t)$ is zero and the diffusion term $\sigma(\boldsymbol{x}, t)$ is $\sigma^2 \mathbf{I}$. We can rewrite equation 5 as

$$\begin{cases} \frac{\partial u(\boldsymbol{x}, t)}{\partial t} = \sigma^2 \Delta u(\boldsymbol{x}, t), x \in \mathbb{R}^d, t \in [0, T] \\ u(\boldsymbol{x}, 0) = f_{\boldsymbol{w}}(\boldsymbol{x}) \end{cases} \tag{12}$$

where $\sigma^2$ is a hyperparameter. In this model, we set the natural trained ResNet as the initial value. Instead of calculating the solution of equation 12, we train a DNN $g_{\boldsymbol{w}}(\boldsymbol{x}, t)$ to approximate the real solution. The parameters of new DNN $g_{\boldsymbol{w}}(\boldsymbol{x}, t)$ are different from the initial ResNet $f_{\boldsymbol{w}}(\boldsymbol{x})$. Therefore, to train the DNN model $g_{\boldsymbol{w}}(\boldsymbol{x}, t)$, we design loss terms on training set $S_N$ for the initial condition and differential equation.

We use $l^{CE}(g_{\boldsymbol{w}}(\boldsymbol{x}_i, 0), f_{\boldsymbol{w}}(\boldsymbol{x}_i))$ to fit the initial condition of equation 12, where $l^{CE}$ is cross-entropy loss. To improve the natural accuracy, we enhance the terminal value to fit the label, i.e., $l^{CE}(g_{\boldsymbol{w}}(\boldsymbol{x}_i, T), y_i)$. We can write the first loss term as

$$L_1(g_{\boldsymbol{w}}, S_N) = \frac{1}{N} \sum_{i=1}^{N} [l^{CE}(g_{\boldsymbol{w}}(\boldsymbol{x}_i, 0), f_{\boldsymbol{w}}(\boldsymbol{x}_i)) + l^{CE}(g_{\boldsymbol{w}}(\boldsymbol{x}_i, T), y_i)]. \tag{13}$$

We use mean square error loss to fit the differential equation and improve the robustness of our model,

$$L_2(g_{\boldsymbol{w}}, S_N) = \frac{1}{N} \sum_{i=1}^{N} \int_0^T \left( \frac{\partial g_{\boldsymbol{w}}}{\partial t}(\boldsymbol{x}_i, s) - \sigma^2 \Delta g_{\boldsymbol{w}}(\boldsymbol{x}_i, s) \right)^2 \mathrm{d}s. \tag{14}$$

According to the trapezoid formula, the integral term equation 14 is approximated by

$$L_2(g_{\boldsymbol{w}}, S_N) \approx \frac{1}{2N} \sum_{i=1}^{N} \sum_{s=0,T} \left( \frac{\partial g_{\boldsymbol{w}}}{\partial t}(\boldsymbol{x}_i, s) - \sigma^2 \Delta g_{\boldsymbol{w}}(\boldsymbol{x}_i, s) \right)^2. \tag{15}$$

Using Taylor formula, the $\Delta g_{\boldsymbol{w}}(\boldsymbol{x}_i, s)$ can be approximated as

$$\Delta g_{\boldsymbol{w}}(\boldsymbol{x}_i, s) \approx \frac{1}{M} \sum_{j=1}^{M} \left( \frac{g_{\boldsymbol{w}}(\boldsymbol{x}_i + h\boldsymbol{\varepsilon}_{i,j}, s) + g_{\boldsymbol{w}}(\boldsymbol{x}_i - h\boldsymbol{\varepsilon}_{i,j}, s) - 2g_{\boldsymbol{w}}(\boldsymbol{x}_i, s)}{h^2} \right). \tag{16}$$

where $\{\boldsymbol{\varepsilon}_{i,j}\}$ is i.i.d and $M$ is the average number. In practice, we choose the number of average $M$ equals 1 to reduce the computation cost, and set $h$ to 0.1 because training converges with difficulty when $h$ is too small. We denote the right hand of equation 16 by $\Delta_{h,\boldsymbol{\varepsilon}} g_{\boldsymbol{w}}(\boldsymbol{x}_i, s)$. Combining equation 15 and equation 16, the integral term equation 14 can be approximated by

$$L_2^{h,M}(g_{\boldsymbol{w}}, S_N) = \frac{1}{N} \sum_{i=1}^{N} \sum_{s=0,T} \left( \frac{\partial g_{\boldsymbol{w}}}{\partial t}(\boldsymbol{x}_i, 0) - \sigma^2 \Delta_{h,\boldsymbol{\varepsilon}} g_{\boldsymbol{w}}(\boldsymbol{x}_i, s) \right)^2. \tag{17}$$

Putting the two objective functions together, our training loss combines $L_1(g_{\boldsymbol{w}}, S_N)$ and $L_2^{h,M}(g_{\boldsymbol{w}}, S_N)$, i.e., $\mathrm{Loss}(g_{\boldsymbol{w}}, S_N) = L_1(g_{\boldsymbol{w}}, S_N) + \lambda L_2^{h,M}(g_{\boldsymbol{w}}, S_N)$, where $\lambda > 0$ is a coefficient to trade off the two loss terms. In practice, we use stochastic gradient descent to optimize $g_{\boldsymbol{w}}(\boldsymbol{x}, t)$, and we use $g_{\boldsymbol{w}}(\boldsymbol{x}, T)$ as the final model.

## 4 EXPERIMENTS

In this section, we will numerically verify that 1) can our method improve the robustness and reduce overfitting risk of DNN? 2) what is the advantage of our method over the Gaussian noise injection and randomized smoothing?

### 4.1 PRELIMINARIES

**Datasets.** In our experiments, we consider both Half-moon and CIFAR10/CIFAR100 (Krizhevsky et al., 2009) datasets. Half-moon dataset is a randomly generated 2d synthetic dataset in which we randomly generate 500 points and 1000 points with a standard deviation of 0.3 as training set and testing set, respectively. Both CIFAR10 and CIFAR100 contain 60K $32 \times 32$ color images with 50K and 10K of them used for training and test, respectively. CIFAR10 and CIFAR100 contain 10 and 100 different classes, respectively.

**Project gradient descent attack (PGD).** To verify the efficiency of our method for improving the robustness of DNN, we consider the project gradient descent (PGD)(Madry et al., 2017) in all the experiments below. Before introducing PGD, we first introduce fast gradient sign method attack (FGSM). The FGSM attack searches the adversarial image $\boldsymbol{x}'$ by maximizing the linearized loss function function $\mathcal{L}(\boldsymbol{x}', y) \approx \mathcal{L}(\boldsymbol{x}, y) + \nabla\mathcal{L}(\boldsymbol{x}, y)^T(\boldsymbol{x}' - \boldsymbol{x})$ with constraint $\|\boldsymbol{x}' - \boldsymbol{x}\|_p \leq \epsilon$, where $\epsilon$ is the maximum perturbation and $p = 2, \infty$. PGD iterates FGSM with step size $\alpha$ and clips the perturbed image to generate the enhanced adversarial attack,

$$\ell^{\infty} : \boldsymbol{x}^{(m)} = \mathrm{Clip}_{\boldsymbol{x}, \epsilon}\{\boldsymbol{x}^{(m-1)} + \alpha \mathrm{sign}(\nabla\mathcal{L}(\boldsymbol{x}^{(m-1)}, y))\},$$

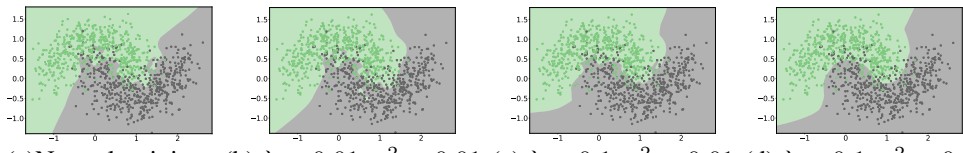

(a)Natural training    (b) $\lambda = 0.01, \sigma^2 = 0.01$   (c) $\lambda = 0.1, \sigma^2 = 0.01$   (d) $\lambda = 0.1, \sigma^2 = 0.1$

**Figure 1:** Decision boundary of natural trained ResNet and ResNet trained by our method with different hyperparameters $\lambda$ and $\sigma^2$.

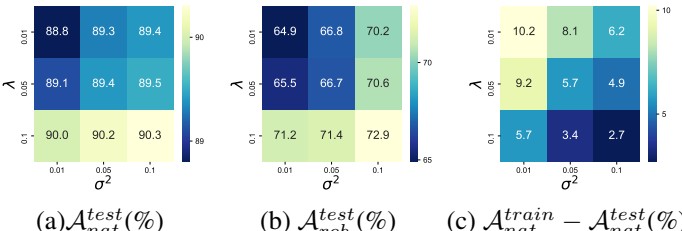

(a)$\mathcal{A}_{nat}^{test}(\%)$      (b) $\mathcal{A}_{rob}^{test}(\%)$      (c) $\mathcal{A}_{nat}^{train} - \mathcal{A}_{nat}^{test}(\%)$

**Figure 2:** Performance of ResNet trained by our model with different hyperparameters $\lambda$ and $\sigma^2$ on Half-moon dataset.

$$\ell^2 : \boldsymbol{x}^{(m)} = \text{Clip}_{\boldsymbol{x},\epsilon}\{\boldsymbol{x}^{(m-1)} + \alpha(\frac{\nabla\mathcal{L}(\boldsymbol{x}^{(m-1)}, y)}{\|\nabla\mathcal{L}(\boldsymbol{x}^{(m-1)}, y)\|_2})\},$$

where $m = 1, \cdots, K$, $\boldsymbol{x}^{(0)} = \boldsymbol{x}$, and let the adversarial image be $\boldsymbol{x}' = \boldsymbol{x}^{(K)}$ with $K$ being the total number of iterations. For the Half-moon dataset, we apply 20 iterations PGD attack (PGD$^{20}$) with step size $0.01$. The perturbation is constrained to be $0.2$ under $l^\infty$ norm. For the CIFAR-10/CIFAR-100 dataset, when under $\ell^\infty$ norm attack, we apply 20 iterations PGD attack with step size $1/255$ and when under $\ell^2$ norm attack and we apply 50 iterations PGD attack (PGD$^{50}$) with step size $0.1$.

**Performance evaluations.** For the Half-moon dataset, we consider three indicators to evaluate the performance of the model: natural testing accuracy ($\mathcal{A}_{nat}^{test}$), robust testing accuracy ($\mathcal{A}_{rob}^{test}$), and the gap between training accuracy and testing accuracy($\mathcal{A}_{nat}^{train} - \mathcal{A}_{nat}^{test}$). The natural accuracy and robust accuracy are measured on clean and adversarial images, respectively. The robust accuracy is used to evaluate the robustness of the models, and the difference between training and testing accuracy is used to evaluate the overfitting risk of the models. For the CIFAR-10/CIFAR-100 datasets, we consider $\mathcal{A}_{nat}^{test}$ and $\mathcal{A}_{rob}^{test}$.

## 4.2 EXPERIMENTS ON SYNTHETIC DATASET

In this subsection, we numerically verify that the efficacy of our training method in improving the robustness and generalization ability of ResNet on the Half-moon dataset.

We first varied hyperparameters $\lambda$ and $\sigma^2$ and present the final results in Figure 2. The three indicators (natural accuracy, robust accuracy, and difference between training accuracy and testing accuracy) for ResNet with natural training were $88.6\%$, $62.3\%$, and $10.4\%$, respectively. Comparing natural trained ResNet, the robustness and generalization ability of ResNet trained with our method were improved. On the other hand, when $\lambda$ and $\sigma^2$ increased, the natural and robust accuracy increased, while the gap between training and test accuracies decreased. Thus we can conclude that increasing $\lambda$ and $\sigma^2$ are helpful for improving the performance of the models.

We plot the decision boundary of ResNet trained with different $\lambda, \sigma^2$ and natural trained ResNet in Figure 1. From Figure 1, we can see that the decision boundary of natural training is irregular, while that of our models is smoother. These experimental results are consistent with our theory.

## 4.3 EXPERIMENT ON CIFAR-10/CIFAR-100 DATASET

We further test the performance of ResNet20/ResNet56 trained with our method on the CIFAR10/CIFAR100 dataset. During training, we apply data augmentation including random crops and flips. We run 200 epochs. The batch size is 128 and the initial learning rate is 0.1, which decays by a factor of 10 at the 100th, and 150th epochs.

| Models | Hyperparameters | $\mathcal{A}_{nat}^{test}$ | $\mathcal{A}_{rob}^{test}$ | | |
|---|---|---|---|---|---|
| | | | $\epsilon = 4/255\,(l^\infty)$ | $\epsilon = 8/255\,(l^\infty)$ | $\epsilon = 0.5\,(l^2)$ |
| **CIFAR-10 Dataset** | | | | | |
| ResNet20 | $\lambda = 0.01, \sigma^2 = 1$ | 86.72% | 40.13% | 7.8% | 45.33% |
| | $\lambda = 0.05, \sigma^2 = 1$ | 84.78% | 45.25% | 13.03% | 50.57% |
| | $\lambda = 0.05, \sigma^2 = 2$ | 82.49% | 48.29% | 17.79% | 53.87% |
| ResNet56 | $\lambda = 0.01, \sigma^2 = 1$ | 87.76% | 41.85% | 8.1% | 46.17% |
| | $\lambda = 0.05, \sigma^2 = 1$ | 86.01% | 47.42% | 14.73% | 53.05% |
| | $\lambda = 0.05, \sigma^2 = 2$ | 83.03% | 48.73% | 18.02% | 55% |
| **CIFAR-100 Dataset** | | | | | |
| ResNet20 | $\lambda = 0.01, \sigma^2 = 1$ | 58.77% | 19.56% | 3.91% | 23.98% |
| | $\lambda = 0.05, \sigma^2 = 1$ | 55.7% | 21.79% | 5.56% | 26.55% |
| | $\lambda = 0.05, \sigma^2 = 2$ | 52.73% | 23.33% | 8.27% | 28.42% |
| ResNet56 | $\lambda = 0.01, \sigma^2 = 1$ | 60.2% | 19.75% | 4.49% | 24.93% |
| | $\lambda = 0.05, \sigma^2 = 1$ | 57.82% | 23.26% | 6.8% | 27.82% |
| | $\lambda = 0.05, \sigma^2 = 2$ | 53.99% | 25.19% | 9.41% | 29.55% |

**Table 1:** Natural and robust accuracies of ResNets trained using our methods with different hyperparameters on the CIFAR-10/CIFAR-100.

| Models | $\mathcal{A}_{nat}^{test}$ | $\mathcal{A}_{rob}^{test}$ | | |
|---|---|---|---|---|
| | | $\epsilon = 4/255\,(l^\infty)$ | $\epsilon = 8/255\,(l^\infty)$ | $\epsilon = 0.5\,(l^2)$ |
| Gaussian noise injection | 89.9% | 22.81% | 12.38% | 0.1% |
| Our ResNet110 | 84.35% | **50.76%** | **19.4%** | **56.38%** |
| ResNet110 | **93.58%** | 0% | 0% | 0.01% |

**Table 2:** Comparing the robustness against $\ell^\infty$-norm and $\ell^2$-norm constrained adversarial perturbations on CIFAR-10.

There are two hyperparameters $\lambda, \sigma^2$ in our algorithm that need to be determined by a large number of experiments. In order to demonstrate the influence of the two hyperparameters, we train ResNet by using our algorithm with different $\lambda, \sigma^2$, then list the result in Table 1. On one hands, from Table 1, we can see that the parameter $\sigma^2$ is fixed, the natural accuracy decreases and robust accuracy increases as the increasing of parameters $\lambda$ and the converse to be true. On the other hand, comparing to the result of ResNet20 and ResNet56, we can see that the performance of our proposed method will be better in deeper networks.

In order to quantitatively measure the performance of our proposed method, we compare the proposed method with other defensive method without adversarial training. The result of comparison is shown in Table 2. For our method, we choose $\sigma^2 = 2$ and $\lambda = 0.5$ to train ResNet110. For the Gaussian noise injection mentioned in subsection 2.2, the standard error of the Gaussian noise equals to 0.5. The experiment result shows that our method is more resistant to adversarial attack than other methods.

## 5 CONCLUDING REMARKS

In this paper, we derived a general form of PDE that can correspond to a DNN model. Motivated by PDE theory, we proposed a training method for ResNet. We have theoretically shown that using our training methods, we obtain a model with better robustness and lower overfitting risk. Based on these theoretical results, we developed an algorithm to train DNN models, and we verified them through experiments. There are numerous avenues for future work: 1) Scale our training method to networks that are large and expressive enough to solve problems like ImageNet. 2) Study other more complex equations for DNN models and solve more challenging real tasks.

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

APPENDIX

## A   PROOF OF THEOREM 1

*Proof.* (Alvarez et al., 1993) The proof of Theorem 1 consists of several steps. First we set

$$\delta_{t,s}(f) = \frac{T_t(f) - T_s(f)}{t - s}, \ \delta_t(f) = \delta_{t,0}(f).$$

Using **[Temporal Regularity]**, for all $h, t \in [0, T]$ and all $f, g \in Q$, we can get

$$\|\delta_t(f + hg) - \delta_t(f)\|_{L^\infty} \le Ch. \tag{18}$$

**Step 1.** We begin the proof of Theorem 1 by proving the following conclusion

**(A1)** the functions $\delta_t(f)$ are Lipschitz continuous uniformly for $t \in (0, 1]$(and uniformly for $f$

belonging to a set $Q$).
Let $\boldsymbol{z}$ be in $\mathbb{R}^d$ and $\|\boldsymbol{z}\|_2 = 1$, we wish to bound $\|\tau_{h\boldsymbol{z}}[\delta_t(f)] - \delta_t(f)\|_{L^\infty}$ for $h \in [0, 1]$. According to **[Spatial Regularity]**, for $f \in Q$, we have

$$\begin{aligned}
&\|\tau_{h\boldsymbol{z}}[\delta_t(f)] - \delta_t(f)\|_{L^\infty} \\
\le \ &\|\tau_{h\boldsymbol{z}}[\delta_t(f)] - \delta_t[f(\boldsymbol{x} + h\boldsymbol{z})]\|_{L^\infty} + \|\delta_t[f(\boldsymbol{x} + h\boldsymbol{z})] - \delta_t[f(\boldsymbol{x})]\|_{L^\infty} \\
\le \ &Ch.
\end{aligned}$$

Then the conclusion of **(A1)** is proved.
**Step 2.** From the preceding steps, we can already deduce a compactness property for $\delta_t(f)$ as $t \to 0$. Now we need more, since we want the whole sequence to converge to the same limit. So we need a Cauchy estimate. Our next step is to prove the following conclusion.

**(A2)** For all $t, s$ in $[0, \frac{1}{2}]$ we have $\|\delta_{t+s,t}(f) - \delta_s(f)\|_{L^\infty} \le m(t)$, where $m(t)$ is some continuous,

nonnegative, nondecreasing function on $[0, \frac{1}{2}]$ such that $m(0) = 0$, and $m(t)$ depends only on bounds

of derivatives of $f$.
First, we can rewrite $\delta_s(f)$ as $\delta_s(f) * K_\varepsilon - [\delta_s(f) * K_\varepsilon - \delta_s(f)]$, where $K \ge 0$, $\int_{\mathbb{R}^d} K\mathrm{d}\boldsymbol{y} = 1$, $K \in C_0^\infty(\mathbb{R}^d)$ and $K_\varepsilon = \varepsilon^{-d} K(\cdot/\varepsilon)$ and $*$ denote the convolution. Using the conclusion of **(A1)**, we can obtain for all $\varepsilon > 0$, there exist a positive constant $C_1$ such that

$$\|\delta_s(f) * K_\varepsilon - \delta_s(f)\|_\infty \le C_1\varepsilon. \tag{19}$$

Then because of **[Markov Property]**, **[Temporal Regularity]** and **[Comparison Principle]**, we have

$$\begin{aligned}
&\|\frac{T_{t+s}(f) - T_t(f + s\delta_s * K_\varepsilon)}{s}\|_{L^\infty} \\
\le \ &\|\frac{T_{t+s,s} \circ T_s(f) - T_t \circ T_s(f)}{s}\|_{L^\infty} + \|\frac{T_t \circ T_s(f) - T_t(f + s\delta_s * K_\varepsilon)}{s}\|_{L^\infty} \\
\le \ &Ct + \|\frac{T_s(f) - f - s\delta_s * K_\varepsilon}{s}\|_{L^\infty} \\
\le \ &Ct + C_2\varepsilon
\end{aligned} \tag{20}$$

On the other hand, since $\delta_s(f) * K_\varepsilon \in C_b^\infty$, **[Temporal Regularity]** implies that

$$\|T_t(f + s\delta_s(f) * K_\varepsilon) - T_t(f) - s\delta_s(f) * K_\varepsilon\|_{L^\infty} \le C_\varepsilon st \tag{21}$$

for some positive constant $C_\varepsilon$ depending only on $\varepsilon$. Combining equation 19, equation 20 and equation 21, we finally deduce that

$$\|\delta_{s+t,t}(f) - \delta_s(f)\|_\infty \le 2C_1\varepsilon + C_\varepsilon t.$$

The **(A2)** holds by setting $m(t) = \inf_{\varepsilon \in (0,1]}(2C_1\varepsilon + C_\varepsilon t)$.
**Step 3.** Now we give a Cauchy estimate in $0 < h \le t \le \frac{1}{2}$ for $\delta_s(f)$

$$\|\delta_t(f) - \delta_h(f)\|_{L^\infty} \le 2\frac{C_0 r}{t} + m(t) \quad \text{where } r = t - [\frac{t}{h}]h. \tag{22}$$

Observe that

$$\delta_t(f) = \frac{Nh}{t}\delta_{Nh}(f) + \frac{r}{t}\delta_{Nh+r,Nh}(f).$$

Using **(A2)** with $s = r$, $t = Nh$ and let $N = [\frac{t}{h}]$, we have

$$\|\delta_{Nh+r}(f) - \frac{Nh}{Nh+r}\delta_{Nh}(f) - \frac{r}{Nh+r}\delta_r(f)\|_{L^\infty} \le \frac{r}{Nh+r}m(Nh) \tag{23}$$

Notice that

$$\delta_{Nh}(f) = \frac{(N-1)h}{Nh}\delta_{Nh}(f) + \frac{h}{Nh}\delta_{Nh,(N-1)h}(f).$$

Using **(A2)** with $s = h$, $t = (N-1)h$, we have

$$\|\delta_{Nh}(f) - \frac{N-1}{N}\delta_{(N-1)h}(f) - \frac{1}{N}\delta_h(f)\|_{L^\infty} \le \frac{1}{N}m((N-1)h) \tag{24}$$

Combining this inequality with equation 24, we obtain

$$\|\delta_t(f) - (N-1)\frac{h}{t}\delta_{(N-1)h}(f) - \frac{h}{t}\delta_h(f) - \frac{r}{t}\delta_r(f)\|_{L^\infty}$$
$$\le \frac{r}{t}m(Nh) + \frac{h}{t}m((N-1)h) \tag{25}$$

Reiterating the argument which lends to equation 24, we obtain after $(N-1)$ more steps that

$$\|\delta_t(f) - \frac{Nh}{t}\delta_h(f) - \frac{r}{t}\delta_r(f)\|_\infty \le \frac{r}{t}m(Nh) + \frac{h}{t}\sum_{j=1}^{N-1}m(jh) \tag{26}$$

Since $m(t)$ is nondecreasing and equation 25, we deduce

$$\begin{aligned}\|\delta_t(f) - \delta_h(f)\|_{L^\infty} &\le& \|\delta_t(f) - \frac{Nh}{t}\delta_h(f) - \frac{r}{t}\delta_r(f)\|_{L^\infty} + \frac{r}{t}\|\delta_h(f) - \delta_r(f)\|_{L^\infty}\\ &\le& \frac{r}{t}m(Nh) + \frac{(N-1)h}{t}m(t) + \frac{r}{t}(\|\delta_h(f)\|_{L^\infty} + \|\delta_r(f)\|_{L^\infty})\\ &\le& m(t) + 2\frac{C_0 r}{t}\end{aligned} \tag{27}$$

Because of equation 27, we can pick $h_n$ going to 0 and $\delta_{h_n}(f)$ converges uniformly on compact sets to a bounded Lipschitz function on $\mathbb{R}^d$ which we denote by $A[f]$

$$\lim_{n\to\infty}\|\delta_t(f) - \delta_{h_n}(f)\|_{L^\infty} \le \lim_{n\to\infty}m(t) + 2\frac{C_0 r}{t}$$
$$\Rightarrow \|\delta_t(f) - A[f]\|_{L^\infty} \le m(t).$$

So $\delta_t(f)$ converges uniformly to $A[f]$ when $t$ goes to 0. Similarly, there exist an operator $A_t$ such that $\delta_{s,t}(f)$ converges uniformly to $A_t[f]$ when $s$ goes to $t$.
**Step 4.** The goal of this step is to show the expression of $A[f]$.
Let $f, g \in C_b^\infty$ and satisfy $f(\mathbf{0}) = g(\mathbf{0}) = 0$ (if not equal to 0, we replace $f(\boldsymbol{x}), g(\boldsymbol{x})$ by $f(\boldsymbol{x}) - f(\mathbf{0}), g(\boldsymbol{x}) - g(\mathbf{0})$), $Df(\mathbf{0}) = Dg(\mathbf{0}) = \boldsymbol{p} \in \mathbb{R}^d$, $D^2 f(\mathbf{0}) = D^2 g(\mathbf{0}) = \mathbf{A} \in \mathbb{R}^{d\times d}$. Then we set $f^\varepsilon = f + \varepsilon\|\boldsymbol{x}\|_2^2$. Using Taylor formula, there exist a positive constant $c$ such that for $\|\boldsymbol{x}\|_2 \le c\varepsilon$ we have $f^\varepsilon \ge g$. Let $w_\varepsilon = w(\boldsymbol{x}/\varepsilon)$ where $w \in C_b^\infty(\mathbb{R}^d)$ and $0 \le w \le 1$ on $\mathbb{R}^d$,

$$\begin{cases} w = 0 & \|\boldsymbol{x}\|_2 \le c/2 \\ w = 1 & \|\boldsymbol{x}\|_2 \ge c \end{cases} \tag{28}$$

Then we introduce $f_0^\varepsilon = w_\varepsilon f^\varepsilon + (1 - w_\varepsilon)g$ and it is obviously that $f_0^\varepsilon \ge g$. Because of **[Comparison Principle]**, $T_t(f_0^\varepsilon) \ge T_t(g)$. Due to

$$f_0^\varepsilon(\mathbf{0}) = f^\varepsilon(\mathbf{0}) = f(\mathbf{0}) = g(\mathbf{0}) = 0 \tag{29}$$

we can also get

$$A(f_0^\varepsilon)(\mathbf{0}) \ge A(g)(\mathbf{0}), A(f^\varepsilon)(\mathbf{0}) \ge A(g)(\mathbf{0}) \tag{30}$$

Because there exists a neighborhood of 0 that $f_0^\varepsilon = f^\varepsilon$, we have $D^\alpha f_0^\varepsilon(\mathbf{0}) = D^\alpha f^\varepsilon(\mathbf{0})$ for $\forall |\alpha| \geq 0$. In view of [Locality] we have $A(f_0^\varepsilon)(\mathbf{0}) = A(f^\varepsilon)(\mathbf{0})$. And considering the continuity of $A$, we can deduce $A(f_0^\varepsilon)$ converges to $A(f)$ in $L^\infty$ when $\varepsilon$ goes to 0. This means $A(f)(\mathbf{0}) \geq A(g)(\mathbf{0})$. By similar method, we can get $A(f)(\mathbf{0}) \leq A(g)(0)$ which means $A(f)(\mathbf{0}) = A(g)(\mathbf{0})$. Hence, for any $\boldsymbol{x}_0 \in \mathbb{R}^d$, if we replace $\mathbf{0}$ by $\boldsymbol{x}_0$, the similar result can be obtained. So the value of $A(f)(\boldsymbol{x})$ only depends on $\boldsymbol{x}, f, Df, D^2 f$. Observe that for any constant $C$, $A(f + C) = A(f)$, we can get $A(f)(\boldsymbol{x})$ only depends on $\boldsymbol{x}, Df, D^2 f$. At last, we finally get

$$A(f) = F(Df, D^2 f, \boldsymbol{x}). \tag{31}$$

Here $F$ is a continuous function. In the same way, we can get

$$A_t(f) = F(Df, D^2 f, \boldsymbol{x}, t). \tag{32}$$

According to **[Ensemble Additivity]**, since

$$F(Df, D^2 f, \boldsymbol{x}, t) = \lim_{t \to 0} (T_t f - f)/t,$$

$F$ therefore satisfies

$$F(rDf + sDg, rD^2 f + sD^2 g, \boldsymbol{x}, t) = rF(Df, D^2 f, \boldsymbol{x}, t) + sF(Dg, D^2 g, \boldsymbol{x}, t)$$

for any real numbers $r$ and $s$ and any functions $f$ and $g$ and at any point $(\boldsymbol{x}, t)$. Since the values of $Df, Dg, D^2 f, D^2 g$ are arbitrary and can be independently taken to be 0, we obtain for any vectors $v_1, v_2$ and symmetric matrices $A_1, A_2$ and any fixed point $(\boldsymbol{x}_0, t_0)$ that

$$F(rv_1 + sv_2, rA_1 + sA_2, \boldsymbol{x}_0, t_0) = rF(v_1, A_1, \boldsymbol{x}_0, t_0) + sF(v_2, A_2, \boldsymbol{x}_0, t_0)$$
$$F(v_1, A_1, \boldsymbol{x}_0, t_0) = F(v_1, 0, \boldsymbol{x}_0, t_0) + F(0, A_1, \boldsymbol{x}_0, t_0).$$

Thus we can finally there exist Lipschitz continuous functions $v : \mathbb{R}^d \times [0, T] \to \mathbb{R}^d$ and Lipschitz continuous positive definite functions $\sigma : \mathbb{R}^d \times [0, T] \to \mathbb{R}^{d \times d}$ such that $u(\boldsymbol{x}, t) = T_t(f)$ is the solution of the equation

$$\begin{cases} \frac{\partial u(\boldsymbol{x}, t)}{\partial t} = v(\boldsymbol{x}, t) \cdot \nabla u(\boldsymbol{x}, t) + \sum_{i,j} \sigma_{i,j}(\boldsymbol{x}, t) \frac{\partial u}{x_i, x_j}(\boldsymbol{x}, t), \boldsymbol{x} \in \mathbb{R}^d, t \in [0, 1] \\ u(\boldsymbol{x}, 0) = f(\boldsymbol{x}), \end{cases}$$

where $\sigma_{i,j}(\boldsymbol{x}, t)$ is the $i, j$-th element of matrix function $\sigma(\boldsymbol{x}, t)$. □

The existence and uniqueness of equation 5 are guaranteed by the following theorem

**Theorem 4.** *(Kolmogorov's backward equation)(Theorem 8.1.1 in (Oksendal, 2013)) Assume Itô process $\boldsymbol{x}(t)$ satisfies SDE*

$$d\boldsymbol{x}(t) = v(\boldsymbol{x}(t), t)dt + \sigma(\boldsymbol{x}(t))d\mathbf{B}(t), \quad \boldsymbol{x}(0) = \boldsymbol{x},$$

*where $\mathbf{B}(t)$ is d-m Brownian motion and $v : \mathbb{R}^d \to \mathbb{R}^d$, $\sigma : \mathbb{R}^d \to \mathbb{R}^{d \times d}$ are all Lipschitz continuous function. Let $f_{\boldsymbol{w}}(\boldsymbol{x}) \in C_0^2(\mathbb{R}^d)$. Then $u(\boldsymbol{x}, t) = \mathbb{E}[f_{\boldsymbol{w}}(\boldsymbol{x}(t)|\boldsymbol{x}(0) = \boldsymbol{x}]$ is the unique solution of equation 5.*

## B  PROOF OF THEOREM 2

To begin the proof of Theorem 2, we first provide the maximum principle for operator $L$ which has the following form

$$Lu = \frac{\partial u}{\partial t} - \sigma^2 \Delta u + v(\boldsymbol{x}, t)\nabla u,$$

where the coefficients $v(\boldsymbol{x}, t)$ are continuous.

**Theorem 5.** *(Evans, 1998) (Maximum principle) Assume $U$ to be an open bounded subset of $\mathbb{R}^n$ and as before set $U_T = U \times (0, T]$ for some fixed time $T > 0$. Let $u \in C^2(U_T) \cap C(\bar{U}_T)$ and $\bar{U}_T$ is the closure of $U_T$. Denote $\bar{U}_T - U_T$ by $\Gamma_T$. If*

$$Lu \leq 0 \text{ in } U_T,$$

*then*

$$\max_{\bar{U}_T} u = \max_{\Gamma_T} u.$$

After the introduction of maximum principle, we next illustrate the proof of the theorem 2.

*Proof.* Let $w(\boldsymbol{x}, t) = (\mu u^2(\boldsymbol{x}, t) + \|\nabla u(\boldsymbol{x}, t)\|^2)e^{-2\lambda t}$, where $\mu$ and $\lambda$ are constants which will be defined later.
Note that $u^2(\boldsymbol{x}, t)$ satisfies

$$\frac{\partial(u^2)}{\partial t} + v(\boldsymbol{x}, t)\nabla(u^2) = \sigma^2 \Delta(u^2) - 2\sigma^2 \|\nabla u\|^2,$$

and $\|\nabla u\|^2$ satisfies

$$\frac{\partial \|\nabla u\|^2}{\partial t} + v(\boldsymbol{x}, t)\nabla \|\nabla u\|^2 = -2\nabla u \nabla v(\boldsymbol{x}, t)\nabla u + \sigma^2 \Delta \|\nabla u\|^2 - 2\sigma^2 (\Delta u)^2,$$

therefore,

$$\frac{\partial w}{\partial t} + v(\boldsymbol{x}, t)\nabla w - \sigma^2 \Delta w$$
$$= e^{-2\lambda t}[-2\lambda(\mu u^2 + \|\nabla u\|^2) - 2\mu\sigma^2 \|\nabla u\|^2 - 2\nabla u \nabla v(\boldsymbol{x}, t)\nabla u - 2\sigma^2(\Delta u)^2].$$

Next, let $\gamma(\boldsymbol{x}, t) = \min_{\|\xi\|=1} \xi^T \nabla v(\boldsymbol{x}, t)\xi$ and $\gamma = \min_{\boldsymbol{x}, t} \gamma(\boldsymbol{x}, t)$, and let $Lw := \frac{\partial w}{\partial t} + v(\boldsymbol{x}, t)\nabla w - \sigma^2 \Delta w$ then we have

$$Lw \leq -2e^{-2\lambda t}[\lambda\mu u^2 + (\lambda + \mu\sigma^2 - \gamma)\|\nabla u\|^2].$$

If we choose $\lambda$ and $\mu$ large enough, such that $\lambda + \mu\sigma - \gamma \geq 0$, then

$$Lw \leq 0.$$

From the maximum principle (Theorem 5), we know $\max_{\boldsymbol{x}} w(\boldsymbol{x}, 1) \leq \max_{\boldsymbol{x}} w(\boldsymbol{x}, 0)$. Let $\mu = 1$ and $\lambda = \gamma - \sigma^2$, we can easily get

$$\|u(\boldsymbol{x}, T)\|_{C^1} \leq e^{-\sigma^2 T} e^{\gamma T} \|f_{\boldsymbol{w}}(\boldsymbol{x})\|_{C^1}, \tag{33}$$

According to the Taylor formula and Hölder inequality, we can get that for any $\|\boldsymbol{\delta}\|_p = \delta$, we have

$$|f_{\boldsymbol{w}}(\boldsymbol{x}) - f_{\boldsymbol{w}}(\boldsymbol{x} + \boldsymbol{\delta})| \leq \|\nabla f_{\boldsymbol{w}}(\boldsymbol{x})\|_q \delta \leq d^{1/q} \|f_{\boldsymbol{w}}(\boldsymbol{x})\|_{C^1} \delta, \tag{34}$$

and

$$|u(\boldsymbol{x}, T) - u(\boldsymbol{x} + \boldsymbol{\delta}, T)| \leq \|\nabla u(\boldsymbol{x}, T)\|_q \delta \leq d^{1/q} \|u(\boldsymbol{x}, T)\|_{C^1} \delta, \tag{35}$$

According to equation 34 and equation 35, we can get

$$u^{c_A}(\boldsymbol{x} + \boldsymbol{\delta}, T) \geq p_A - d^{1/q} \|u(\boldsymbol{x}, T)\|_{C^1} \delta$$
$$\max_{i \neq c_A} u^i(\boldsymbol{x} + \boldsymbol{\delta}, T) \leq p_B + d^{1/q} \|u(\boldsymbol{x}, T)\|_{C^1} \delta.$$

Thus $u^{c_A}(\boldsymbol{x} + \boldsymbol{\delta}, T) \geq \max_{i \neq c_A} u^i(\boldsymbol{x} + \boldsymbol{\delta}, T)$ for all

$$\delta \leq \frac{p_A - p_B}{2d^{1/q} \|u(\boldsymbol{x}, T)\|_{C^1}}$$

Then combining equation 33, equation 34 and equation 35 we can easily prove the theorem. $\square$

## C    PROOF OF THEOREM 3

The goal of the learning problem is to find $h \in \mathcal{H}$ such that **population risk** $E(h) := \mathbb{E}_{(\boldsymbol{x}, y)}[\ell(h(\boldsymbol{x}), y)]$ is minimized. We consider the supervised learning setting where one has access to $n$ i.i.d. training examples $S_N$. A learning algorithm maps the $N$ training examples to a hypothesis $h \in \mathcal{H}$. In this section, we are interested in the gap between the **empirical risk** $E_N(h) := \frac{1}{N}\sum_{i=1}^{N} \ell(h(\boldsymbol{x}_i), y_i)$ and the population risk $E(h)$, known as the generalization error. We start with the following theorem which connects the population and empirical risks via Rademacher complexity.

**Lemma 1.** *(Theorem 3.5 in (Mohri et al., 2018)) Let $S_N = \{\boldsymbol{x}_i, y_i\}_{i=1}^N$ be samples chosen i.i.d. according to the distribution $p(\boldsymbol{x})$. If the loss function $\ell$ is bounded by $B > 0$. Then for any $\delta \in (0, 1)$, with probability at least $1 - \delta$, the following holds for all $h \in \mathcal{H}$,*

$$E(h) \leq E_{S_N}(h) + 2R_{S_N}(\ell_{\mathcal{H}}) + 3B\sqrt{(\log(2/\delta)/(2N))}.$$

*In addition, according to the Ledoux-Talagrand contraction inequality(Maurer, 2016) and assume loss function is L-Lipschitz, we have*

$$R_{S_N}(\ell_{\mathcal{H}}) \leq LR_{S_N}(\mathcal{H}).$$

Next, we begin to prove Theorem 3.

**Theorem 6.** *(Theorem 2.3 in (Mendelson, 2003)) Let $\mathcal{H}$ be a class of functions from $\mathcal{D}$ to $[-1, 1]$ and set $p$ to be a probability measure on $\mathcal{D}$. Let $(\boldsymbol{x}_i)_{i=1}^n$ be independent random variables distributed according to $\mu$. Let $(Y, d)$ be a metric space and set $F \subset Y$. For every $\epsilon > 0$ and any $n \geq 8/\epsilon^2$,*

$$\mathbb{P}\left(\sup_{f \in \mathcal{H}} |\frac{1}{n}\sum_{i=1}^n f(\boldsymbol{x}_i) - \int_{\mathcal{D}} f(\boldsymbol{x})\mu(\boldsymbol{x})\mathrm{d}\boldsymbol{x}| > \epsilon\right) \leq 8\mathbb{E}_\mu[\mathrm{Num}(\epsilon/8, \mathcal{F}, L_1(\mu_n))]\exp(-n\epsilon^2/128) \quad (36)$$

In above theorem, $\mathrm{Num}(\epsilon, \mathcal{H}, L_p(\mu_n))$ denotes the covering numbers of $\mathcal{H}$ at scale $\epsilon$ with respect to the $L_p(\mu_n)$ norm. $\mu_n$ is the empirical measure supported on one sample of $(\boldsymbol{x}_i)_{i=1}^n$. For every $\epsilon > 0$, denote by $\mathrm{Num}(\epsilon, \mathcal{H}, d)$ the minimal number of open balls (with respect to the metric $d$) needed to cover $\mathcal{H}$. That is, the minimal cardinality of the set $\{y_1, \cdots, y_m\} \subset Y$ with the property that every $f \in \mathcal{H}$ has some $y_i$ such that $d(f, y_i) < \epsilon$. The set $\{y_1, \cdots, y_m\}$ is called an $\epsilon$-cover of $\mathcal{H}$ . The logarithm of the covering numbers is called the entropy of the set. For every sample $\{\boldsymbol{x}_1, \cdots, \boldsymbol{x}_n\}$ let $\mu_n$ be the empirical measure supported on that sample. For $1 \leq p < \infty$ and a function $f$, $\|f\|_{L_p(\mu_n)} = \left(\frac{1}{n}\sum_{i=1}^n |f(\boldsymbol{x}_i)|^p\right)^{1/p}$ and $\|f\|_\infty = \max_{1 \leq i \leq n} |f(\boldsymbol{x}_i)|$.

Notice that

$$L_1(\mu_n) \leq L_\infty(\mu_n) \leq L_\infty.$$

We get one immediate corollary of Theorem 6.

**Corollary 1.** *Let $\mathcal{H}$ be a class of functions from $\mathcal{D}$ to $[-1, 1]$ and set $p(\boldsymbol{x})$ to be a probability measure on $\mathcal{D}$. Let $\{\boldsymbol{x}_i\}_{i=1}^N$ be independent random variables distributed according to $p(\boldsymbol{x})$. For every $\epsilon > 0$ and any $n \geq 8/\epsilon^2$,*

$$\mathbb{P}\left(\sup_{f \in \mathcal{H}} |\frac{1}{N}\sum_{i=1}^n f(\boldsymbol{x}_i) - \int_{\mathcal{D}} f(\boldsymbol{x})p(\boldsymbol{x})\mathrm{d}\boldsymbol{x}| > \epsilon\right) \leq 8\mathrm{Num}(\epsilon/8, \mathcal{H}, L_\infty)\exp(-N\epsilon^2/128) \quad (37)$$

*where $\mathrm{Num}(\epsilon, \mathcal{H}, L_\infty)$ be the covering numbers of $\mathcal{H}$ at scale $\epsilon$ with respect to the $L_\infty$ norm*

Then, we get an upper bound of $\sup_{f \in \mathcal{H}} |\frac{1}{N}\sum_{i=1}^N f(\boldsymbol{x}_i) - \int_{\mathcal{D}} f(\boldsymbol{x})p(\boldsymbol{x})\mathrm{d}\boldsymbol{x}|$.

**Corollary 2.** *Let $\mathcal{H}$ be a class of functions from $\mathcal{D}$ to $[-1, 1]$. Let $\{\boldsymbol{x}_i\}_{i=1}^N$ be independent random variables distributed according to $p$, where $p$ is the probability distribution whose $C^1$ norm is bounded. Then with probability at least $1 - \delta$,*

$$\sup_{f \in \mathcal{H}} |p(f) - p_N(f)| \leq \sqrt{\frac{128}{N}\left(\ln \mathrm{Num}(\sqrt{\frac{2}{N}}, \mathcal{H}, L_\infty) + \ln\frac{8}{\delta}\right)},$$

*where*

$$p(f) = \int_{\mathcal{D}} f(\boldsymbol{x})p(\boldsymbol{x})\mathrm{d}\boldsymbol{x}, \quad p_N(f) = \frac{1}{N}\sum_{i=1}^N f(\boldsymbol{x}_i). \quad (38)$$

*Proof.* Using Corollary 1, with probability at least $1 - \delta$,

$$\sup_{f \in \mathcal{H}} |p(f) - p_N(f)| \leq \epsilon_\delta,$$

where $\epsilon_\delta$ is determined by

$$\epsilon_\delta = \sqrt{\frac{128}{N}\left(\ln \text{Num}(\epsilon_\delta/8, \mathcal{H}, L_\infty) + \ln\frac{8}{\delta}\right)}.$$

Obviously,

$$\epsilon_\delta \geq \sqrt{\frac{128}{N}} = 8\sqrt{\frac{2}{N}}$$

which gives that

$$\text{Num}(\epsilon_\delta/8, \mathcal{H}, L_\infty) \leq \text{Num}(\sqrt{\frac{2}{N}}, \mathcal{H}, L_\infty)$$

Then, we have

$$\epsilon_\delta \leq \sqrt{\frac{128}{N}\left(\ln \text{Num}(\sqrt{\frac{2}{N}}, \mathcal{H}, L_\infty) + \ln\frac{8}{\delta}\right)}$$

which proves the corollary. $\qquad\square$

If the entropy bound of $\mathcal{H}$ is known, the upper bound of $\sup_{f\in\mathcal{H}}|p(f) - p_N(f)|$ follows from Corollary 2. Now, the key point left becomes bounding the entropy of some given function class $\mathcal{H}$.

Let us start from the function class $\mathcal{F}$. To apply above corollary, we need to normalize $\mathcal{F}$ to make it lie in $[-1, 1]$. Here we also use $\mathcal{F}$ to denote the normalized function class and absorb the bound of $\mathcal{F}$ into the generic constant $C$.

According to Taylor formula, we have for any $\boldsymbol{x}, \boldsymbol{z} \in \mathcal{D}$

$$|f(\boldsymbol{x}) - f(\boldsymbol{z})| \leq \sqrt{d}\sup_{f\in\mathcal{F}}\|f\|_{C^1}\|\boldsymbol{x} - \boldsymbol{z}\|_2.$$

Where $d$ is the dimension of $\boldsymbol{x}$. This gives an easy bound of $\text{Num}(\epsilon, \mathcal{F}, L_\infty)$,

$$\text{Num}(\epsilon, \mathcal{F}, L_\infty) \leq C\left(\frac{\sup_{f\in\mathcal{F}}\|f\|_{C^1}}{\epsilon}\right)^d \tag{39}$$

Using Corollary 2, with probability at least $1 - 1/(2N)$, we have

$$\sup_{f\in\mathcal{F}}|p(f) - p_N(f)| \leq \frac{C}{\sqrt{N}}\left(\ln N + \ln(\sup_{f\in\mathcal{F}}\|f\|_{C^1}) + 1\right)^{1/2} \tag{40}$$

and

$$\sup_{f\in\mathcal{G}_\sigma}|p(f) - p_N(f)| \leq \frac{C}{\sqrt{N}}\left(\ln N + \ln(\sup_{f\in\mathcal{G}_\sigma}\|f\|_{C^1}) + 1\right)^{1/2}$$

According to equation 33, we can get the following theorem.

**Theorem 7.** *With probability at least* $1 - 1/(2N)$,

$$\sup_{f\in\mathcal{F}}|p(f) - p_N(f)| \leq \frac{C}{\sqrt{N}}\left(\ln N + \ln c + 1\right)^{1/2},$$

$$\sup_{f\in\mathcal{G}_\sigma}|p(f) - p_N(f)| \leq \frac{C}{\sqrt{N}}\left(\ln N + \ln c - \sigma^2 + 1\right)^{1/2}$$

$$p(f) = \int_{\mathcal{D}} f(\boldsymbol{x})p(\boldsymbol{x})\mathrm{d}\boldsymbol{x}, \quad p_N(f) = \frac{1}{N}\sum_{\boldsymbol{x}_i\in S_N} f(\boldsymbol{x}_i),$$

*$\mathcal{F}$ and $\mathcal{G}_\sigma$ is a function class defined as*

$$\mathcal{F} := \Big\{f(\boldsymbol{x})|\|f\|_{C^1} \leq c\Big\},$$

*and*

$$\mathcal{G}_\sigma := \Big\{g(\boldsymbol{x}) = u(\boldsymbol{x}, 1)|\frac{\partial u(\boldsymbol{x}, t)}{\partial t} = v(\boldsymbol{x}, t)\cdot\nabla u(\boldsymbol{x}, t) + 1/2\sigma^2\Delta u(\boldsymbol{x}, t), u(\boldsymbol{x}, 0) = f(\boldsymbol{x}) \text{ where } f\in\mathcal{F}\Big\}.$$

For the purpose of proving Theorem 3, we give a inequality about Rademacher complexity as following.

**Theorem 8.** *(Theorem 2.23 in (Mendelson, 2003)) Let $p(\boldsymbol{x})$ be a probability distribution function and set $\mathcal{H}$ to be a class of functions on $\mathcal{D}$. Denote*

$$Z = \sup_{f \in \mathcal{H}} |\frac{1}{N} \sum_{i=1}^{N} f(\boldsymbol{x}_i) - \mathbb{E}_{\boldsymbol{x} \sim p(\boldsymbol{x})}(f)|,$$

*where $\{\boldsymbol{x}_i\}_{i=1}^{N}$ are independent random variables distributed according to $p(\boldsymbol{x})$. Then we have*

$$\frac{1}{2}\mathbb{E}_{\boldsymbol{x} \sim p(\boldsymbol{x})}Z \leq R_{S_N}(\mathcal{H}) \leq 2\mathbb{E}_{\boldsymbol{x} \sim p(\boldsymbol{x})}Z + \frac{1}{\sqrt{N}}|\sup_{f \in \mathcal{H}} \mathbb{E}_{\boldsymbol{x} \sim p(\boldsymbol{x})}f|.$$

Then combining Theorem 7, Theorem 8, we have

$$R_{S_N}(\mathcal{G}_\sigma) \leq 2\mathbb{E}_{\boldsymbol{x} \sim p(\boldsymbol{x})}Z + \frac{1}{\sqrt{n}}|\sup_{f \in \mathcal{G}_\sigma} \mathbb{E}_{\boldsymbol{x} \sim p(\boldsymbol{x})}f|.$$

$$\leq \frac{C}{\sqrt{N}} \left(\ln N + \ln c - \sigma^2 + 1\right)^{1/2} + \frac{1}{\sqrt{N}}|\sup_{f \in \mathcal{G}_\sigma} \mathbb{E}_{\boldsymbol{x} \sim p(\boldsymbol{x})}f|$$

$$\leq \frac{C}{\sqrt{N}} \left(\ln N + \ln c - \sigma^2 + 1\right)^{1/2} + \frac{1}{\sqrt{N}}\sup_{f \in \mathcal{G}_\sigma} \|f\|_{C^1}$$

$$\leq \frac{C}{\sqrt{N}}[(-\sigma^2 + \ln c + \ln N)^{1/2} + \exp(-\sigma^2)c].$$

Similarly, we can estimate the upper bound of $R_{S_N}(\mathcal{F})$.

## D    MORE EXPERIMENTAL DETAILS ON HALF-MOON DATASET

In this subsection, we plot the decision boundary of ResNet trained with different hyperparameters in Figure 3. As drawn in Figure 1, we can see as the hyperparameters $\lambda$ and $\sigma^2$ increasing, the decision boundary becomes more smooth, which implies the overfitting risk reduce. These experimental results are also consistent with the conclusion of Theorem 3.

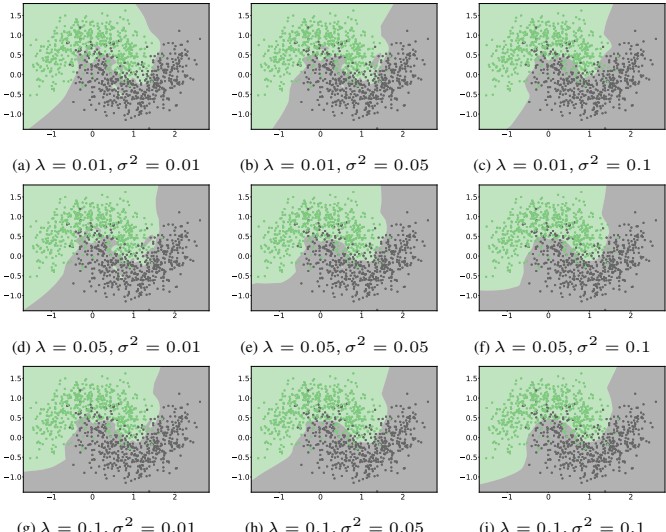

(a) $\lambda = 0.01, \sigma^2 = 0.01$     (b) $\lambda = 0.01, \sigma^2 = 0.05$     (c) $\lambda = 0.01, \sigma^2 = 0.1$

(d) $\lambda = 0.05, \sigma^2 = 0.01$     (e) $\lambda = 0.05, \sigma^2 = 0.05$     (f) $\lambda = 0.05, \sigma^2 = 0.1$

(g) $\lambda = 0.1, \sigma^2 = 0.01$     (h) $\lambda = 0.1, \sigma^2 = 0.05$     (i) $\lambda = 0.1, \sigma^2 = 0.1$

**Figure 3:** Boundary decision of ResNet trained by our model with different hyper parameters $\lambda$ and $\sigma^2$.

## E    ARCHITECTURES OF THE USED DNNS

Figure 4 shows the architectures of ResNets used in this paper. We also plot the inputs for the ResNet used on CIFAR-10 dataset in Figure 5 and every element of the matrix $tt$ equals to $t$.

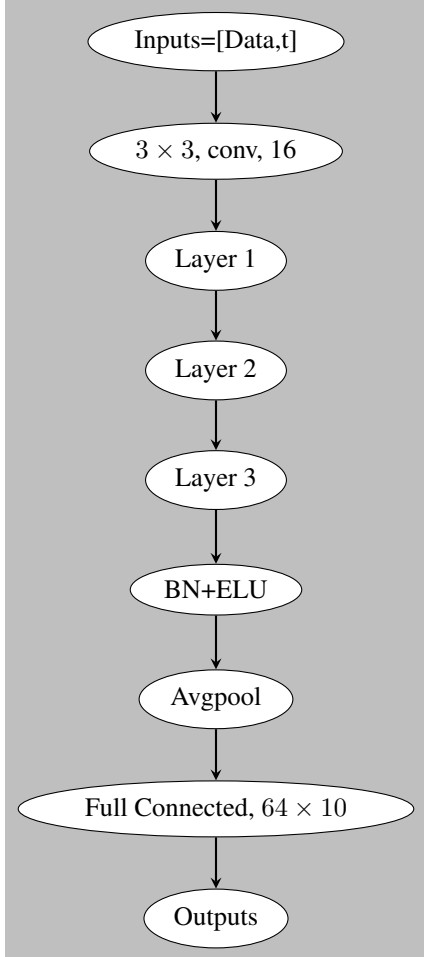

**Figure 4:** Architectures of the ResNet used in our experiments.

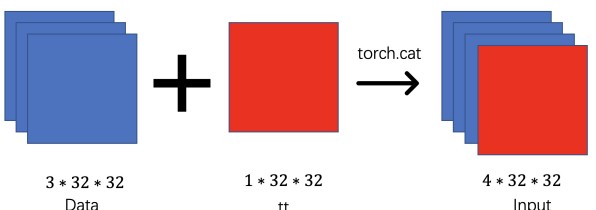

**Figure 5:** The inputs for the ResNet used in CIFAR-10 dataset.

