# OpenReview forum: "A framework of deep neural networks via the solution operator of partial differential equations"
_ICLR.cc/2022/Conference — ICLR 2022 Submitted_

### Official Review · Reviewer_ZDUG · 2021-11-02

**Correctness:** 3
**Technical Novelty And Significance:** 3
**Empirical Novelty And Significance:** 3
**Recommendation:** 6
**Confidence:** 3

**Details Of Ethics Concerns:**

N.A.

**Main Review:**

# Post-rebuttal Comments (11/29/2021)

I thank the authors for their responses and updating the manuscript. The updated version improved the readability. Since my questions are clarified, I want to keep my score (6).

# Initial Comments
【Strength】
- [1] This paper provided an architecture-independent sufficient condition that a convection-diffusion PDE models a DNN.
- [2] The experiments are consistent with the implications of theoretical analyses (assuming that theorems are true. See【Correctness】section)

【Weakness】
- [1] Theorem 2 and Theorem 3 apply to restricted PDE (7) (constant-coefficient $\sigma$) instead of the general PDE (1) derived in Theorem 1.
- [2] Although I tried my best, I could not confirm the correctness of some parts of the proof. I would suggest writing their details in more detail.

【Correctness】
- [1] P.7: In experiments in Section 4, how is the initial ResNet $f_w$ trained?
- [2] P.15: I could not confirm the correctness of the proof of Theorem 2. Specifically, I want to know the following things (I guess some of them came from the idea of [Wang et al., 2020]. But I could not figure out):
  - [2-1] the precise statement of the maximum principle (for parabolic operators) used in the proof
  - [2-2] how we can derive (31) from the conditions $Lw\leq 0$ and $\max_x w(x, 1)\leq \max_x w(x, 0)$
  - [2-3] how we can derive the statement from (31)--(33).
- [3] P.18: I think it is better to write how to derive the bounds for $\mathcal{G}_\sigma$ in Theorem 6 from (the proof of) Theorem 2. More specifically, the bounds come from (31) in the proof of Theorem 2.
- [4] P.18: It is not trivial to me how to derive Theorem 3 from the conclusions of Theorem 6 and Theorem 7

【Technical Novelty And Significance】
- [1] Many studies pointed out that ResNet and its variants can be described by discretizing ODEs (e.g., Neural ODE [Chen et al., 2018]). Also, [Wang et al., 2019] linked these models to PDEs. However, this paper investigated the "opposite" direction. It derived the condition under which a model is can be described by a PDE (Theorem 1). I think this is novel, as to the best of my knowledge, I do not know a general theory about it.
- [2] Also, I think this paper is significant as it gives a new guideline for creating new deep learning models. Also, it provides a hint with how we go beyond the regime of the convection-diffusion models.
- [3] I think the proof of Theorem 2 is based on the idea of [Wang et al. 2020]. On the other hand, it is a new idea to derive the bound for Rademacher complexity of $\mathcal{G}_\sigma$ from this idea.

【Empirical Novelty And Significance】
- [1] If I understand correctly, the proposed model is a discretization of the same PDE as the Gaussian noise injection model. However, the proposed model is novel as the training method differs from Gaussian noise injection.
- [2] In addition, according to Table 2, the proposed method can increase the robustness against adversarial attacks. Therefore, I think this study is empirically significant.

【Detailed Comments】
- [1] P.1: Without loss of generality, we indicate this evolutive relationship by an adjustment operator, ... → I could not understand what "Without loss of generality" meant in this context. I would like to clarify it.
- [2] P.3: ... Half-moon and CIFAR-10/CIFAR-100 dataset. → ... datasets.
- [3] P.3: $S_N\in \mathcal{D}$　→ $\mathcal{S}_N\subset \mathcal{D}$
- [4] P.3: [Markov property] → [Markov Property]
- [5] P.3: [Spatial Regularity]: $\tau_hf$ → $\tau_{\mathbfit{h}}f$ (Make $h$ boldface.)
- [6] P.3: [Temporal Regularity]: property → properties
- [7] P.3: [Temporal Regularity]: $C_n$ is undefined.
- [8] P.4: We then require [...] that no enhancement is made. → I could not understand what "no enhancement is made". Could you clarify the meaning?
- [9] P.4: If I understand correctly, this paper claimed without proofs that when $u(x, 0) = \mathrm{softmax} (w^{\mathrm{FC}}x(0))$ and $x(t)$ evolves by $dx(t) = v(x(t), t)dt$, then $u(x, t) = \mathrm{softmax}(w^{\mathrm{FC}}x(t))$ satisfies the PDE (5) (Let me know if I misunderstood what this paper intended). I suggest writing how to derive it (I guess we can think of it as a corollary of Feynman-Kac formula. But since there are no stochastic terms, we may have simpler proof.)
- [10] P.5: $\mathcal{Y}$ is undefined.
- [11] P.5: ... depend on $\nabla v$. → ... depending on $\nabla v$.
- [12] P.6: Suppose $S_N$ ... is drawn from $X\times Y \subset \mathcal{D} \times \{-1, 1\}$ ... Assume $\mathcal{D}$ is the underlying distribution of $X\times Y$. → It seems the usage of $\mathcal{D}$ is ambiguous. $\mathcal{D}$ is a subset of some Euclidean space in the first sentence while $\mathcal{D}$ is the distribution on the subset in the second sentence.
- [13] P.6: It looks weird that the initial ResNet $f_w$ and the trained ResNet $g_w$ has the same parameter $w$ because they are independent networks. Does the parameter $w$ represent the concatenation of learnable parameters of $f_w$ and $g_w$?
- [14] P.6: $l^{CE} = -\log g_k(x)$: Does this equation assume that $f_w(x) = k$, that is, the output of $f_w$ is not a probability distribution over $\{1, \ldots, K\}$ but a one-hot vector.
- [15] P.7:  PGD with step size α and ... the total number of iterations. → I think this sentence is grammartically difficult to read. I would suggest reconsidering the wording.
- [16] P.7: $x'=x^{(K)}$ → $\mathbfit{x}'=\mathbfit{x}^{(K)}$
- [17] P.8: Is there any reason why the experiments did not compute generalization gap for CIFAR-10 and CIFAR-100 datasets?
- [18] P.9: $\lambda.\sigma^2$ → $\lambda, \sigma^2$
- [19] P.9: nature accuracy → natural accuracy
- [20] P.9: showned → shown
- [21] P.13: (Alvarez et al., 1993) → What is this reference for?
- [22] P.13: $\tau_{hz}$ → $\tau_{h\mathbfit{z}}$ (Make $z$ boldface)
- [23] P.13: $Ch+\|\delta_t[f(\mathbfit{x}+h\mathbf{z}) - f(\mathbfit{x})]\|_{L^\infty}$ → Remove the $Ch$ term.
- [24] P.15, (30): I think $A_t(f)$ is undefined.
- [25] P.16, P.18 : I think it is better to write the statements of the maximum principle and Khinchin's inequality as readers may be less familiar with these theorems.

**Summary Of The Paper:**

This paper studied a ResNet-like DNN model that can be expressed as a discretization of PDEs. First, this paper showed that, under some assumptions, any adjust operator is a solution of a second-order convection-diffusion PDE (Theorem 1). ResNets and ResNets with Gaussian noise injection are special cases of this theorem. Next, for the specific PDE (Eq. (7)), this paper derived generalization guarantees regarding the Rademacher complexity. This paper also derived robustness guarantees in terms of input perturbations. Finally, this paper analyzed the predictive performance of ResNet trained by the proposed method with various model hyperparameters on clean and adversarial datasets.

**Summary Of The Review:**

This paper is technically significant as it gave a model-agnostic PDE theory that encompasses several DNN models. I also think this paper is empirically significant as they support the theory well. On the other hand, I could not confirm that some parts of the proofs are correct (see【Correctness】 section). Since they are likely to be minor issues, I scored the correctness rating at 3. However, if these questions cannot be resolved, I may decrease the rating of correctness.

---

> ### Author Response · Authors · 2021-11-21
> **Response to Reviewer ZDUG**
>
> Q1: Theorem 2 and Theorem 3 apply to restricted PDE (7) (constant-coefficient ) instead of the general PDE (1) derived in Theorem 1.
>
> Reply: Compared to the constant diffusion coefficient, it is significantly harder to consider the non-constant diffusion coefficient, and we leave the case of the non-constant diffusion coefficient as future work.
>
> Q2: In experiments in Section 4, how is the initial ResNet $f_w$ trained?
>
> Reply: We train the initial ResNet $f_w$ with natural training whose parameters can be learned by minimized the loss $l^{CE}(y_i,f_{w}(x_i))$, where $l^{CE}$ is cross-entropy loss. And we have supplemented the details in section 3.
>
> Thanks for your consideration for the parts of the proof in this paper. We have revised our manuscript according to your suggestion, and the revised parts are highlighted in blue. We hope we have cleared your concerns about our work. We have also revised our manuscript according to your comments, and we would appreciate it if we can get your further feedback at your earliest convenience.

---

### Official Review · Reviewer_zQvx · 2021-11-02

**Correctness:** 3
**Technical Novelty And Significance:** 2
**Empirical Novelty And Significance:** 2
**Recommendation:** 5
**Confidence:** 4

**Main Review:**

While I like the idea behind this work, I must say that I was disappointed by the way it was presented here.

There are two main reasons:
- The writing is lacking in clarity and thoroughness. For example, the operator T_t, which is the central object in the paper, is never rigorously defined. There is a formula for its discrete counterpart but the continuous one, while mentioned in the introduction, only formally appears in Theorem 1 which meaning thus becomes unclear, and one has to guess that the operator which is being characterized is defined through the properties above. This is only one example, albeit the most striking one. Another one is the explanation of the so-called fundamental properties which I found to be quite unconvincing: after several readings, while I understand why those properties are necessary from a technical point of view for the proof of Theorem 1, I don't see a clear motivation for them defining T_t.
- The contributions of the paper are not clear and neither are its assumptions. For example, in Theorem 2, the base model is supposed to be compactly supported: why should this be the case? It is not true for a linear classifier for example. Moreover, while I found the experiments interesting, the toy dataset experiments were not convincing: is the claim that the proposed method can be used to ensure adversarial robustness? In this case, there should be other baseline methods to compare to.

Update after rebuttal:

I thank the authors for the improvements made in the updated version of the paper. While I still have some doubts regarding the significance of the contribution, I wouldn't be against its acceptance and I raise my score.

**Summary Of The Paper:**

This paper considers DNNs as transformations of the input data which can be seen as discrete approximations to the solutions of PDEs. It reformulates a point of view which has been trending for a few years now. More precisely, this paper represents the network as an operator, which is actually solution to a PDE, which acts on a "base classifier". Using facts from PDE theory, they then propose improvements for standard neural architectures.

**Summary Of The Review:**

While the work is promising, it is not rigorous nor clearly written enough to justify acceptance as there are shortcomings both in terms of the expositions and regarding the substance of the work.

---

> ### Author Response · Authors · 2021-11-21
> **Response to Reviewer zQvx**
>
> Thank you for your valuable feedback and thoughtful reviews. We have revised our manuscript according to your suggestion, and the revised parts are highlighted in blue. Below we address your concerns.
>
> Q1: The operator $T_t$ is never rigorously defined and the explanation of the so-called fundamental properties are quite unconvincing.
>
> Reply: We are sorry for the misunderstanding of the definition of the operator $T_t$. We have modified the introduction of this paper for a more precise definition of the operator $T_t$. Since we hope that operator $T_t$ is the solution operator of PDEs, it is natural that the solution of the PDEs should have continuity and stability in temporal and spatial. So it is necessary for the operator $T_t$ to have these fundamental properties. Using operator $T_t$ is to derive the general form of the PDEs which can correspond to DNN. And our main contribution of this paper to find that the general form of the PDEs that can correspond to the neural network.
>
> Q2: In Theorem 2, the base model is supposed to be compactly supported: why should this be the case?
>
> Reply: We assume the data points are distributed in a bounded domain, and we also assume the function corresponding to a neural networks model is defined in the bounded domain. So the assumption that the base model is supposed to be compactly supported is reasonable.
>
> Q3: The toy dataset experiments were not convincing: is the claim that the proposed method can be used to ensure adversarial robustness? In this case, there should be other baseline methods to compare.
>
> Reply: In fact, we do not only show the results on toy dataset but also show experimental results on CIFAR-10 and CIFAR-100 datasets in section 4.3. We also use Gaussian noise injection ResNet (ResNet Ensemble method without adversarial training proposed in this paper "ResNets Ensemble via the Feynman-Kac Formalism to Improve Natural and Robust Accuracies" by Wang et. al, 2018.) as baseline method. The experiment results show that the robustness of ResNet110 trained with our method is better than Gaussian noise injection ResNet.
>
> ======================================================
>
> We hope we have cleared your concerns about our work. We have also revised our manuscript according to your comments, and we would appreciate it if we can get your further feedback at your earliest convenience.

---

### Official Review · Reviewer_mSDj · 2021-11-03

**Correctness:** 3
**Technical Novelty And Significance:** 2
**Empirical Novelty And Significance:** 2
**Recommendation:** 3
**Confidence:** 4

**Main Review:**

The paper connects resnet architecture with transport equation and a noise injected resnet with a diffusion equation. The framework introduced is certainly not novel, as shown in this paper "ResNets Ensemble via the Feynman-Kac Formalism to Improve Natural and Robust Accuracies" by Wang et. al, 2018. In fact, they too use it as a framework for adversarial robustness, and authors have not compared their methodology with this paper.

Secondly, the algorithm used to retrieve the final network uses a resnet as an initialization, so I suspect that all this famework is doing is approximating a noise injected version of the original resnet, in a rather complicated way.

In general the authors should improve the exposition of the paper as it has a lot of grammatical mistakes and misused words. The paper needs to be revised carefully.

**Summary Of The Paper:**

This paper connects the architecture of a resnet and a gaussian injected resnet to that of a transport equation and a diffusion equation respectively. They show that under this framework they provide robustness guarantees of their PDE framework scale with the $\sigma$ parameter of the diffusion equation. Furthermore, they also show that larger the $\sigma$ better the generalization gap for the resulting neural network.

They introduce a learning algorithm, to attain the network from their PDE framework starting from an initial NN (also a resnet).

They also empirically verify that their method improves robustness, and that it performs better than Gaussian Noise injection.

**Summary Of The Review:**

I think that the framework provided by the authors is not novel enough, and the authors haven't empirically verified their methodology with relevant previous work. Further, the paper in the current state is not very well written.

---

> ### Author Response · Authors · 2021-11-21
> **Response to Reviewer mSDj**
>
> Thanks for handling our submission. We have some concerns about Reviewer mSDj’s comments. We feel that Reviewer mSDj did not read our paper carefully at all. We feel Reviewer mSDj is quite irresponsible, and the comments reflect a serious misunderstanding of our submission. Below we elaborate on this in detail.
>
> First, let us summarize our contribution, which seems ignored or misunderstood by reviewer mSDj. Different from the predecessors who found a specific form of PDE to design the neural network architecture, the main contribution of this paper is to find that the general form of the PDE that can correspond to the neural network is precisely the second-order convection-diffusion equation. The coefficient of this diffusion term can be a non-constant function. Second, neither our framework nor the practical algorithm proposed uses a technique of noise injection. Using Gaussian noise in our practical algorithm (section 3) is to approximate the Laplace calculator, which is essentially different from the goal of adding noise for robustness. Moreover in section 2.2 of this paper, we use the Feynman-Kac formula to illustrate noise injected ResNets model of the paper "ResNets Ensemble via the Feynman-Kac Formalism to Improve Natural and Robust Accuracies" is a special case of our framework. We compare robustness against adversarial perturbation of our method and ResNets Ensemble without adversarial training (proposed in the paper "ResNets Ensemble via the Feynman-Kac Formalism to Improve Natural and Robust Accuracies" by Wang et. al, 2018, and called this method Gaussian noise injection) in section 4.3 Table 2. The experimental results show that the robustness of ResNet110 trained with our method is better than Gaussian noise injection ResNets.
>
> At last, we emphasize the point again that this framework is not approximating a noise injected version of the original ResNets in a complicated way and the goal of this paper is to find the general form of the PDE that can correspond to the neural network. The proposed training method is to train a ResNet to approximate the solution of PDE.
>
> We are sorry for grammatical mistakes and misused words and we will attempt to improve the exposition of the paper. Thanks for your consideration. Please let us know if any further clarification is needed.

---

### Official Review · Reviewer_LaXY · 2021-11-04

**Correctness:** 4
**Technical Novelty And Significance:** 3
**Empirical Novelty And Significance:** 2
**Recommendation:** 6
**Confidence:** 4

**Main Review:**

Apart from a few typos, the paper is well-written and generally easy to follow and understand.
The proofs, while not containing any novel techniques, are rigorous and correct. Viewing a
DNN model as the flow of a regular Markovian operator seems quite natural and generalizes
the neural ODE idea. I also appreciate that the authors derive an explicit PDE based on
very reasonable assumptions on the desired properties of a generic classifier. I also really
like the illustrative example for ResNet and the Gaussian model injection, essentially
viewing neural ODE(s) as the characteristic curves of a transport equation which is a special
case of their more general convection-diffusion equation (for ResNet). I also liked the
robustness guarantee which serves as an inspiration for using this framework to build models
that are better adept at handling adversarial examples. Generally, I think stability
is the right motivation for considering such frameworks.

The algorithms and numerics are the weaker parts of this work. It is unclear why the
authors choose to reduce their PDE (1) to a re-scaled heat equation after showing that
ResNet(s), models that are known to work well, come from a transport equation. Perhaps
this simply makes the PDE easier to solve with less parameters to be learned. Nevertheless,
I think exploring the more general form should certainly be done given the first part
of the paper places a lot of emphasis on this result. My guess as to why the methods
ends up working is that the diffusion form the heat equation is able to smooth out the loss function
which results in finding more stable regions of parameter space (perhaps there is a
connection to https://arxiv.org/pdf/1704.04932.pdf which takes a similar point of
view but with a viscous Hamilton-Jacobi equation on SGD instead)? The method of
solving this heat equation just with another neural network seems a bit uninspired
and is simply a special case of PINNS (https://www.sciencedirect.com/science/article/pii/S0021999118307125).
While the PDE is very high dimensional, it is also very simple so applying some
Using classical techniques to solve it will, I think, make for a very interesting
comparison. Furthermore the numerics are not nearly extensive enough (I do appreciate
the half moon example showing the smoothed decision boundary). The authors need
to consider more network architectures, not just ResNet, as well as harder datasets,
like ImageNet, and more types of adversarial attacks to really make a case that this
method is practically useful. Furthermore, there seems to be a significant decrease
in accuracy for clean images. If this is to be practical, this issue must be resolved
as sacrificing 10% in accuracy for some robustness is usually not acceptable.

**Summary Of The Paper:**

The authors cast DNN classifiers as the push-forward of a base classifier under a
flow map at some fixed final time. Under some reasonable assumptions on the flow,
they show that, given any base classifier, the flow map can be obtained as the solution
to a convection-diffusion equation. They show that ResNets and Gaussian noise injection
can be viewed as special cases of their model and give a robustness guarantee for any
classifier defined as a solution to their PDE. Experiments on the 2-d half moon data set
as well CIFAR 10 and 100 show better robustness  of their model to adversarial attacks
when compared to standard ResNet(s).

**Summary Of The Review:**

Very interesting connection between DNN(s) and PDE(s), generalizing neural ODE(s) but
the practical algorithms presented don't make a strong case for the method.

---

> ### Author Response · Authors · 2021-11-21
> **Response to Reviewer LaXY**
>
> Thank you for your valuable feedback and thoughtful reviews. Below we address your concerns.
>
> Q1: It is unclear why the authors choose to reduce their PDE (1) to a re-scaled heat equation after showing that ResNet(s), models that are known to work well, come from a transport equation.
>
> Reply: Because ResNet can be viewed as the solution of transport equation which uses softmax model as the initial value, as a whole, our model can be regarded as taking the softmax model as the initial value, first solving a transport equation and then solving a heat equation. Thus the velocity $v(t,x)$ can be is involved with the transport equation (ResNet). Therefore, our model only needs to take ResNet as the initial value and solve the solution of a heat equation.
>
> Q2: The authors need to consider more network architectures, not just ResNet, as well as harder datasets, like ImageNet, and more types of adversarial attacks to really make a case that this method is practically useful.
>
> Reply: Because our device cannot support large datasets, we can not verify the performance of our training method on ImageNet, and we will leave it as future work. We also use C&W2, FGSM and other attacks to verify the robustness of our model, but the PGD attack is more effective than other attack methods, so we only list the experimental results of the PGD attack.
>
> Q3: There seems to be a significant decrease in accuracy for clean images.
>
> Reply: In fact, according to the experiment results, we can find the nature accuracy decreases and robust accuracy increases as the increasing of parameters $\sigma^2$ and $\lambda$. So we only need to choose smaller parameters $\sigma^2$ and $\lambda$ for higher natural accuracy.
>
> ===============================================
>
> We hope we have cleared your concerns about our work. We have also revised our manuscript according to your comments, and we would appreciate it if we can get your further feedback at your earliest convenience.

---

### Decision · Program_Chairs · 2022-01-20

**Decision:**

Reject

**Comment:**

The paper formulates ResNet like classifiers as the evolution of a base classifier through an operator corresponding to a PDE, up to a given final time. Using a set of assumptions on the desired properties of the flow operator, the authors show that it can be obtained as the solution of a convection-diffusion equation. This generalizes ideas developed e.g. for Neural ODEs. The authors provide several examples showing that their formulation encompasses regularization methods proposed for deep NNs.  They further provide robustness guaranties for a classifier defined according to their framework. They introduce an algorithm based on a restricted version of their framework and propose different experiments showing the increased robustness of their model to a family of adversarial attacks compared to baseline ResNets.

The paper introduces an original idea, providing a very interesting connection between ResNets and PDEs. This allows the authors to exploit known properties of PDEs and opens the way to new theoretical insights on DNNs while allowing the development of DNN models with proved properties. As mentioned this generalizes the view of ResNets introduced in Neural ODEs.
Besides, the paper presents weaknesses. First the form will make it accessible only to a very small audience in the ML community. No effort is made in the writing to introduce the required PDE concepts that would help a lot understanding and appreciating the contribution. This is a pity since given the current trend on this topic this could be of interest to a large community. Then the use cases in the experiments focus solely on robustness properties and one type of attacks. This illustrates only one aspect of the potential of the framework, and this does not provide a strong case in support of the ideas introduced before. The global message carried out by the paper then becomes unclear. Overall, the current version could be largely improved and this will certainly lead to a strong contribution.